# Characterization of Electrospun BDMC-Loaded PLA Nanofibers with Drug Delivery Function and Anti-Inflammatory Activity

**DOI:** 10.3390/ijms241210340

**Published:** 2023-06-19

**Authors:** María José Morillo-Bargues, Andrea Olivos Osorno, Consuelo Guerri, Manuel Monleón Pradas, Cristina Martínez-Ramos

**Affiliations:** 1Center for Biomaterials and Tissue Engineering, Universitat Politècnica de València, Cno. de Vera s/n, 46022 Valencia, Spain; mamobar2@upvnet.upv.es (M.J.M.-B.); mmonleon@ter.upv.es (M.M.P.); 2Departamento de Ingeniería Biomédica, Universidad Iberoamericana, Prolongación Paseo de la Reforma 880, Lomas de Santa Fe, Ciudad de México 01219, Mexico; 3Molecular and Cellular Pathology of Alcohol Laboratory, Prince Felipe Research Institute, 3 Eduardo Primo Yúfera Street, 46012 Valencia, Spain; cguerri@cipf.es; 4Biomedical Research Networking Center in Bioengineering Biomaterials and Nanomedicine (CIBER-BBN), 28029 Madrid, Spain; 5Department of Medicine, Universitat Jaume I, Av. Vicent-Sos Baynat s/n, 12071 Castellón de la Plana, Spain

**Keywords:** bisdemethoxycurcumin, Schwann cells, poly-L-lactic acid, electrospinning, inflammasome

## Abstract

Controlled drug release systems are the subject of many investigations to achieve the therapeutic effect of drugs. They have numerous advantages, such as localized effects, lower side effects, and less onset of action. Among drug-delivery systems, electrospinning is a versatile and cost-effective method for biomedical applications. Furthermore, electrospun nanofibers are promising as drug carrier candidates due to their properties that mimic the extracellular matrix. In this work, electrospun fibers were made of Poly-L-lactic acid (PLA), one of the most widely tested materials, which has excellent biocompatible and biodegradable properties. A curcuminoid, bisdemethoxycurcumin (BDMC) was added in order to complete the drug delivery system. The PLA/BDMC membranes were characterized, and biological characteristics were examined in vitro. The results show that the average fiber diameter was reduced with the drug, which was mainly released during the first 24 h by a diffusion mechanism. It was seen that the use of our membranes loaded with BDMC enhanced the rate of proliferation in Schwann cells, the main peripheral neuroglial cells, and modulated inflammation by reducing NLRP3 inflammasome activation. Considering the results, the prepared PLA/BDMC membranes hold great potential for being used in tissue engineering applications.

## 1. Introduction

The use of electrospun (e-spun) fibers as a functionalized system for controlled drug delivery has numerous advantages compared to conventional dosage forms, such as improved therapeutic effect, reduced toxicity, convenience, and so on [1,2]. Poly-L-lactic acid (PLA) is one of the most versatile and widely tested materials that has been approved by the US Food and Drug Administration (FDA) to be used in direct contact and with biological fluids [3]. Curcumin is known due to its potential antioxidant, anti-inflammatory, and antimicrobial properties [4].

Electrospinning is an effective method already demonstrated to produce ultrafine fibers of several materials with different diameter ranges and large surface areas [5,6]. These parameters make them ideal for the mimic of a natural extracellular matrix required for tissue engineering (TE), with promoted cell adhesion, migration, and proliferation [3,7,8]. The possibility of large-scale production combined with the simplicity of the process makes this technique very attractive for many different applications [9]. Electrospinning assembly can be modified in different ways for combining material properties [6]. Therefore, there is an ever-increasing interest in this field of electrospinning because of the useful properties of the e-spun fibers, which are suitable for biomedical applications [5,6].

PLA is a thermoplastic biopolymer whose precursor molecule is lactic acid. PLA is known as one of the most promising biopolymers for TE due to its biodegradability, mechanical properties, and biocompatibility [10,11]. Bisdemethoxycurcumin (BDMC) is a component of turmeric, which is a spice that comes from the root Curcuma longa and is a flowering plant of the ginger family. The use of turmeric dates back nearly 4000 years but interest in curcumin continues to grow as evidenced by the numerous bibliographic reviews [12,13,14].

The yellow color of turmeric is mainly due to the presence of curcuminoids, which are hydrophobic polyphenols with poor solubility in water. The major curcuminoids are curcumin, demethoxycurcumin (DMC), bisdemethoxycurcumin, and the most recently discovered cyclocurcumin (CYC), also called curcumin-I, curcumin-II, curcumin-III, and curcumin IV, respectively [15]. However, the term “curcumin” is confusingly also used in the literature and in commercial applications to describe the curcumin extract that contains all four curcuminoids (I, II, III, and IV) at different concentrations [16,17].

Curcuminoids share similar anti-inflammatory properties, but they differ in some aspects [17]. BDMC shows higher antimetastasis potency than cur-I [18,19] and has an inhibitory activity significantly greater than that of curcumin-I and curcumin-II over some leukemia cell lines (K562, KBM-5) [20] and MAPK mitogen- and stress-activated kinase protein, which promotes the synthesis of inflammatory cytokines [16]. Further, BDMC is chemically more stable than Cur-I and DMC [21,22] due to the absence of methoxy groups on its phenyl moieties. Contrary to that, DMC and Cur-I were found to possess one and two methoxy groups, respectively [19,23,24]. As to the rates of alkaline degradation, BDMC values are lower than Cur-1 and DMC. Cur-I is stable at acid pH because of the conjugated diene structure. In neutral or basic conditions, the phenolic OH is deprotonated and it becomes unstable [25].

Recently, efforts have been made to develop an effective drug delivery system for curcumin based on an electrospinning technique for biomedical applications [26,27,28]. Due to its pharmacologic properties, curcumin may be a potential treatment for peripheral nerve injury (PNI) [29,30] and other tissue damages. It is known that, after PNI, inflammatory events occur. Inflammation is a defense mechanism of the organism against harmful stimuli, such as infection, antigen challenge, or tissue injury [31]. Moderate inflammatory response is a key to host defense, and it helps to repair damaged tissues. However, uncontrolled inflammation may cause the opposite effect [32].

Inflammasomes are cytosolic multiprotein complexes that are responsible for the inflammatory response of innate immunity [33]. The most extensively studied is the NLRP3 inflammasome or cryopyrin, which is formed by the cytosolic sensor molecule NLRP3, the adaptor protein ASC, and the effector molecule pro-caspase-1 [34,35]. NLRP3 promotes inflammation by increasing IL-18 and IL-1β synthesis in a two-step model, priming and activation step [36,37]. Lipopolysaccharide (LPS) found in the outer membrane of Gram-negative bacteria can act in the first step of priming. LPS induces inflammation processes via toll-like receptor 4, and, consequently, NF-kβ is activated by phosphorylation, leading to the transcription of pro-inflammatory cytokines genes such as tumor necrosis factor-alpha (TNF-α) and interleukin 1β (IL-1β). LPS acts as a pathogen-associated molecular pattern (PAMP). For the second signal, exogenous ATP can be used to induce NLRP3 assembly and mature IL-18 and IL-1β release. ATP present in living cells is released from dying and stressed cells and may act as a damage-associated molecular pattern (DAMP) [38].

The purpose of this work is to obtain a BDMC release system and study the effects of this curcuminoid in isolation. The compartmentalized information will help us to understand their specific pharmacological roles and to improve the selective usage of curcumin and its curcuminoids. In this work, we prepared PLA/BDMC composite membranes by electrospinning at two different BDMC concentrations and evaluated their physical properties and in vitro behavior. Toward this aim, we fabricated e-spun PLA-BDMC-loaded fibers to obtain a functionalized system for gradual drug release. This may be promising in the future for TE, for neural repair therapies, using a combination of living cells, engineering materials, and biochemical factors [39].

## 2. Results

### 2.1. Synthesis and Morphology of the E-Spun Nanofiber

Figure 1A–C shows the macroscopic aspect of three representative fabricated e-spun membranes. The yellow color increased according to the proportion of curcuminoid as expected. The morphology of these e-spun fibers was characterized and is shown in Figure 1D–I. It can be observed that the surface of the fibers was smooth and uniform, without the presence of any beads. E-spun fibers were aligned due to the use of a rotating disc as a collector. Figure 1 also shows that morphology was maintained for all samples, which were cylindrical-shaped, but the average diameter of the e-spun fibers loaded with different amounts of BDMC decreased [7,26]. The diameters of the fibers were obtained by measuring 30 random fibers of each image for a total of 9 samples using the Image J software. The diameters of the B1 and B2 fibers were lower than those of PLA in a very significant way. The mean diameter of the PLA nanofibers was estimated to be 0.84 ± 0.28 μm, while for the PLA/BDMC composite membranes, it was 0.35 ± 0.13 for B1 and 0.45 ± 0.17 μm for B2 membrane nanofibers. Comparatively, the BDMC addition to the electrospinning solution also narrowed the diameter distribution. The average of the PLA fibers’ diameter doubled those of the PLA/BDMC composite membranes, suggesting that the amount of BDMC and DMSO present in the electrospinning solution had a significant effect on the final diameter of the electrospun fibers. However, there was no direct correlation between BDMC content and fiber diameter.

### 2.2. Physicochemical Characteristics

#### 2.2.1. FTIR

The chemical structure of the PLA/BDMC composite membranes was studied by FTIR. Figure 2A shows the FTIR spectra of the BDMC and PLA/BDMC composite membranes and the pure PLA membrane. It can be seen that the pure PLA and PLA/BDMC composite membranes had similar spectra.

The FTIR spectrum of PLA fibers and the spectra of the BDMC-loaded samples show small differences in the region between 1700 and 1500 cm^−1^, where bands associated with the carbonyl group and aromatic rings may appear. These rings are typical of curcuminoids and were observed in the PLA/BDMC composite membranes. To be more specific, the peak at 1627 cm^−1^ was a stretching vibration peak of the carbon–carbon double bonds in BDMC. The detection of this peak ensured the existence of BDMC in the composite membrane.

On the other hand, due to the high ratio of PLA/BDMC, typical peaks of PLA were observed clearly and strongly in all membranes, PLA, B1, and B2.

#### 2.2.2. DSC

PLA is a semicrystalline polyester which can undergo crystallization and melting upon heating. It has been shown that the PLA processed by electrospinning increased its crystallinity. The DSC thermograms of the B1, B2, and PLA fibers and BDMC are shown in Figure 3. The peak just above 65 °C in the thermograms corresponds to the glass transition of the amorphous phase of PLA. The exothermic region between 85 °C and 110 °C (approx.) corresponds to crystallization taking place in the amorphous phase once this has become mobile after the glass transition, and finally the endothermic peak around 170 °C corresponds to the melting of the crystalline phase of PLA. The DSC thermograms reveal no specific thermal events for BDMC in this temperature range. The comparison of the curves in Figure 3 reveals that the electrospun samples possessed, as produced, a much smaller crystalline phase than the commercial PLA, and this is due to the rapid evaporation of the solvent upon electrospinning which impedes crystallization. The effect of the BDMC load on this process is to diminish the ability of PLA to crystallize immediately after the glass transition, which translates to less intense endotherms in this range and, correspondingly, a smaller melting peak at a high temperature.

#### 2.2.3. TGA

From the thermograms given in Figure 4, it can be seen that the char residue of BDMC was much higher than that of the others, which should be due to the phenyl rings in BDMC. Thermal decomposition of the curcuminoid below 700 °C occurred in two consecutive steps. The first step was between 237 and 423 °C, with a 50% of mass loss. The second stage began at 423 °C and ended at 700 °C resulting in approximately 15% weight loss. BDMC had a higher temperature of maximum decomposition, and, even if the temperature increased to 100 °C, the BDMC did not degrade more.

Pure BDMC and PLA membranes had a higher onset temperature of degradation compared with PLA/BDMC composite membranes. Pure PLA fibers and the BDMC started to decompose at 248 °C. However, when PLA and BDMC were mixed in the electrospinning solution, there was an obvious influence on the thermal stability of PLA/BDMC films.

Figure 4 also shows that B2 fibers were thermally less stable than B1 and these, in turn, were less stable than those of PLA. The loss of thermal stability of PLA/BDMC composite membranes may be attributed to the smaller diameter of the fibers. The weight loss occurs in the samples with BDMC at lower temperatures. This may be due to the fact that in these samples, the diameter of the nanofibers is smaller and, therefore, the diffusion of volatile products of thermal degradation is easier.

The final residual mass was more abundant as the amount of BDMC present in the fibers increased, which is related to the high thermal stability of BDMC. If we analyzed the percentage weight loss at a selected temperature of 570 °C, it was clearly observed that the residual mass increased in the BDMC-loaded PLA fibers in a directly proportional way to the amount of BDMC they had. We obtained 0.032 and 0.105 as the mass fraction of B1 and B2 membranes, respectively. These results were concordant with the previous analysis, corroborating the higher presence of BDMC in B2.

### 2.3. In Vitro Drug Release Assay

The kinetics of the drug release (Figure 5) from the e-spun devices showed that the maximum value of BDMC in the medium was obtained on the first day. The maximum release of the drug was observed within the first 24 h for both the B1 and B2 membranes, where more than 80% of the BDMC was eluted to the medium. These calculations were made in relation to the total of BDMC released by the membranes submerged in PBST during successive days.

It was also possible to perform the calculations using the total amount of drug loaded in the fibers. In this case, the drug load was 21 µg BDMC/g membrane for B1 and 109 µg BDMC/g membrane for B2. The values of BDMC released in the first 24 h were 7.7 and 9.5 µm for B1 and B2, respectively. Then, considering drug loading efficiency, the percentages of BDMC released in the first 24 h decreased to 40 and 10% for B1 and B2, respectively. In this case, our results showed that the quantity released by B1 quadruples that of the B2 membrane when compared to the amount of drug loaded.

The BDMC diffusion coefficient was obtained from the slope of the release curve and was 1.20 for B2 y 0.90 for B1 membranes.

### 2.4. Cell Viability

The MTS test was performed in SCs cells to evaluate the cell viability of the e-spun PLA/BDMC-loaded membranes. The absorbance, at 490 nm wavelength, was directly proportional to the number of viable cells (Figure 6). According to the results derived from Figure 6, we can see how the B1 effect increased the absorbance values with respect to the control on the 1st and 3rd days, whereas this difference was not significant on day 7.

In the case of B2 membranes, only this increase was observed when compared to the control on day 3, with values very similar to those of B1. However, on day 1, a significant difference was found when comparing the two groups with BDMC between themselves. B1 continued improving cell proliferation compared to B2, but the difference was not as significant as it was for the control. The highest proliferative effect (*p* < 0.001) was observed on day 1 when comparing B1 to control.

### 2.5. Immunocytochemistry

Figure 7A–C provided a representative example of differences in SCs proliferation according to BDMC concentration. The immunocytochemistry assay for the marker S100β, a repeatedly used SCs marker, and DAPI suggested a suitable cell viability of SCs on the B1 membranes. After 24 h of culture, the B1 membranes showed the highest SCs proliferation compared with the control and B2 membranes. This result corroborated the one obtained in the MTS assay.

Figure 7D–F showed, at a higher magnification, the relationship between SCs’ orientation and e-spun nanofiber alignment. When taking a closer look, cells presented elongated shaping due to aligned e-spun fibers providing contact guidance to cultured SCs, influencing their cell morphology. The SCs’ cytoplasm orientation along the direction of the nanofibers was better observed in PLA membranes. This may be due to the treatment with DMSO of the PLA/BDMC composite membranes in order to remove BDMC autofluorescence before immunostaining. The morphological alteration would be more pronounced as the concentration of BDMC increases in the membrane.

### 2.6. RT-qPCR

In order to evaluate the anti-inflammatory potential of our PLA/BDMC composite membranes, we seeded SCs on them. After 24 h, SCs were stimulated in two steps with LPS and ATP inducing an inflammatory response (Figure 8). The results demonstrated that, under pro-inflammatory conditions, the expression of NF-kβ was significantly reduced in the presence of BDMC when compared to other groups. Moreover, TNFα decreased very significantly in the B1 and B2 membranes. The components of the inflammasome complex, NLRP3 and Casp1, had lower expression levels when cells were seeded on membranes loaded with BDMC. The results also showed IL-18 mRNA expression was decreased in presence of BDMC in a highly significant way.

The results demonstrated that, under pro-inflammatory conditions, the expression of NF-kβ was significantly reduced in the presence of BDMC when compared to other groups. Moreover, TNFα decreased significantly in the B1 and B2 membranes. The components of the inflammasome complex, NLRP3 and Casp1, had lower expression levels when cells were seeded on membranes loaded with BDMC. The results also showed that IL-18 mRNA expression was significantly decreased in the presence of BDMC compared to the control.

Consistently, the results from RT-qPCR indicated that the addition of BDMC to the e-spun membranes had suppressive effects on inflammasome activation, assembly, and the subsequent release of the mature form of IL-18 compared to the control in a no-dose-dependent manner.

### 2.7. Inmunoblotting Assays

In concordance with the previous result, the Western blot corroborated the anti-inflammatory effect of the PLA/BDMC composite membranes under pro-inflammatory conditions (Figure 9). The protein expression profiles showed that membranes loaded with BDMC reduced NLRP3 inflammasome-related proteins. NLRP3 protein and Procaspase1, both part of the inflammasome complex, were significantly decreased in B1 and B2 membranes compared to other stimulated groups.

The same occurred with the Caspase1p20 subunit, the active form of caspase-1, and mature IL-18 levels. Our results revealed that the presence of BDMC in different concentrations significantly inhibited proinflammatory proteins’ production.

## 3. Discussion

TE is a scientific research area with a massive increasing interest in recent years for an effective approach to tissue repair [40]. Many research publications are looking for a scaffold that will be able to mimic the biological environment to meet this challenge. The non-woven e-spun nanofibrous membranes provide this promising 3D substrate for cell growth with cost-effective technology and versatility. Their effectiveness has also been demonstrated in a variety of applications such as a drug release mechanism [41,42].

In this work, we synthesized and characterized a composite system formed by aligned PLA nanofibers loaded with BDMC, a structural analog of curcumin, in order to enhance the efficiency of a local application of this drug. The lipophilicity of BDMC makes it very soluble in a PLLA/dichloromethane/dimethylformamide solution and the surface presence of curcuminoid aggregates was not observed. As can be noticed, the incorporation of the drug dissolved in DMSO had a conductivity enhancement effect on the electrospinning PLA solution (conductivity of 0.60 ± 0.04 µS/cm for PLA, 0.77 ± 0.02 µS/cm for B1 membranes and 0.79 ± 0.01 µS/cm for B2 membranes).

The conductivity of the electrospinning solution plays a key role in fiber-forming technology. This may be a consequence of BDMC playing the role of a salt when incorporated into the electrospinning solution. It is known that the addition of salts to the solution increases the conductivity and consequently the electric force for the stretching of the jet, which promotes a reduction in the diameter of the fibers [43,44,45]. It has been found that the increase of the electric charges enhances the jet instability of the electrospinning process, and eventually leads to the increase of fiber diameter polydispersity [7].

Under the conditions used in this study, e-spun fibers exhibited different values of surface area to mass ratio. However, no changes were observed in their surfaces without compromising the bulk properties of the biomaterial. It was established that there were statistically significant differences between the diameter of the PLA/BDMC composite membranes and the unloaded, independent of BDMC content. However, the increased diameter of the B2 fibers in reference to B1 may be the result of variations in the electrospinning solution properties, such as viscosity, due to differences in drug concentration. The studies revealed that an increase in viscosity may be promoted by an increase of a BDMC structural analog, curcumin [45], and the use of DMSO as a drug solvent. The DMSO used for the dissolution of BDMC in the BDMC/PLA composite membranes is also related to fiber diameter variation [46]. DMSO may modify some solution properties such as viscosity, conductivity, and surface tension. These variations might contribute to a decrease in the diameter of the B1 and B2 fibers [47].

The polymer solution with BDMC resulted in a decrease in the average fiber diameter, which lead to lower thermal stability. Pure BDMC had high thermal stability because of its benzene rings. However, when BDMC was added to PLA membranes its thermal stability became worse with the increase of the drug. This fact could be due, as we have seen before, to the increase in the conductivity of the electrospinning solution, which would result in the synthesis of finer e-spun fibers. The smaller diameter of the fibers increased their surface–mass ratio, which would reduce their thermal stability.

A growing number of studies have suggested that curcumin can promote the regeneration and functional recovery of injured peripheral nerves. In the literature, it has been reported that curcumin promotes the migration, differentiation, and proliferation of SCs while decreasing apoptosis [39,48]. In the present study, we evaluated the effect of BDMC, a structural analog of curcumin, on SCs. Our results showed that low doses of BDMC promoted SCs proliferation at short culture times in a very significant way when compared to the control. One day after treatment, B1 membranes showed the highest proliferative effect of the drug. After 3 days of SCs culture, B1 and B2 presented very similar values, and both increased when compared to non-treated cells. When the release kinetic was studied, a fast release of the drug in all cases was observed. This behavior indicated that most of the BDMC was loaded on the surface of the fibers. B1 was still the group with the highest SCs proliferation, but this result was not significant with respect to the other groups. This result may be explained by the proliferative effect of BDMC being no longer visible. The small amount of curcuminoid that was released into the environment after a week of culture had no proliferative power. The drug release was not enough to have an effect on the cells. Most of the drug released in our study was eluted within the first day. The e-spun PLA-BDMC-loaded fibers have a more surface area-to-volume ratio than PLA fibers because of their smaller diameter. For this reason and for the fact that more than 80% of the total drug released was eluted in the first 24 h, it is believed that the BDMC was deposited mostly on the surface of the e-spun fibers.

This fast release of the drug would allow cells to grow, generating a convenient cell layer of SCs on the membranes in less time than the control. Moreover, our PLA/BDMC composite membranes were able to provide contact guidance by the aligned fibers to SCs. It is established that aligned e-spun fibers stimulate SCs maturation and, therefore, trophic support for axonal growth [49]. This fact may be convenient after a PNI because of the need for a rapid response of the SCs to remove the myelin remains and to create suitable support for restoring the nerve to its functional state. It is known that the effect of the aligned substrate is significant on axon and glial cell orientation and maturation [49,50,51]. For instance, following PNI, SCs are essential not only to clean the injured area but also to support nerve regeneration. It is demonstrated that aligned SCs release growth factors that enhance neurite growth and guide the axons [49,51]. Our e-spun fibers may provide optimal support and anchorage for cells and control cell behavior.

Our studies revealed that the B1 membranes released about four times more BDMC compared to B2 membranes when these data were compared with the amount of drug loaded. This may be due to the fact that the B1 fibers had a smaller diameter than B2 and therefore a greater surface to which the drug remained adhered without penetrating the interior of the fiber.

The bioavailability of curcuminoids is limited by their poor water solubility among other properties [52,53,54]. They are promising drug candidates for the treatment of many diseases, but their use in therapeutic treatments is hampered by low aqueous solubility and chemical instability [55]. Achieving a release system of a drug inside the human body such as BDMC, with limited aqueous solubility, is aimed at developing new therapeutic strategies. Due to the electrospinning technique, this hydrophobic small drug, BDMC, more chemically stable than curcumin and DMC and can be loaded for local delivery.

It was observed that BDMC maintained its biomedical functional properties and could be mostly released quickly after tissue damage when inflammation processes are activated. However, there is also a small concentration of BDMC encapsulated into e-spun fibers which will be slowly released as the nanofibers degrade. Therefore, our results indicated that the process of electrospinning does not affect the proliferative and anti-inflammatory activity of the BDMC and showed its biocompatibility.

It is known that the most studied curcuminoid, curcumin, has anti-inflammatory properties [56,57,58] and enhances the rate of proliferation and migration in SCs, the main peripheral neuroglial cells, and reduces SCs apoptosis [48]. The anti-inflammatory effects of curcumin occur in many ways. Curcumin has been shown to suppress NF-kβ transcription factor in inflammation [59,60]. Curcumin also acts by preventing the spread of the inflammation process, blocking the binding between TNF-α and its receptor [60,61]. Additionally, curcumin can suppress the spatial formation of the NLRP3 inflammasome components [56]. Consequently, curcumin treatment significantly prevents inflammatory processes.

In the present study, we found that, in proinflammatory conditions, BDMC acted in a similar way to its analogous curcumin. Our results showed that the mRNA expression of TNF-α, NF-Kβ, Casp1, NLRP3, and IL18 in cultured SCs was decreased in the presence of BDMC-loaded membranes. Further, protein levels of NLRP3 and Casp1, components of the inflammasome protein complex, and the proinflammatory interleukin IL18 were significantly inhibited by BDMC. Accordingly, the PLA/BDMC composite membranes would have an inhibitory effect on the inflammatory response. Specifically, BDMC would suppress the NF-Kβ signaling pathway, preventing NLRP3 subunits’ assembly and the release of proinflammatory cytokines including TNF-α expression. Figure 10 shows the functional structure of the NLRP3 inflammasome to understand the parts that would be compromised by BDMC.

In this work, we synthesized aligned e-spun membranes with a drug encapsulated into the e-spun fibers. In the literature, loading drugs into delivery systems such as liposomes, nanoparticles, emulsions, conjugates, etc., has been developed because of its advantages to improve therapeutic efficacy [61]. In this context, we fabricated e-spun fibers to combine properties such as strength and lightness and a large surface area to volume ratio [62]. The electrospinning process allows for fast and efficient solvent evaporation and offers the chance for loading highly concentrated molecules of BDMC onto the fiber surface. This BDMC maintains its biomedical functional properties and can be released quickly after tissue damage when inflammation processes are activated by diffusion phenomenon. However, there is also BDMC encapsulated into e-spun fibers which will be slowly released as the nanofibers degrades. Overall, the obtained results provide a good acknowledgment that the PLA/BDMC membranes may be a proper release system for tissue engineering applications.

Since both BDMC membranes have similar anti-inflammatory effects, it might be more interesting for further work to use the B1 membrane. This membrane achieves a higher increase in cell proliferation in the short term with half the amount of drug than to B2. In addition, the manufacturing cost would be lower by using smaller amount of BDMC.

## 4. Materials and Methods

### 4.1. Electrospun Fiber Mat on Polymer Membrane Film

To obtain PLA e-spun fibers, the following steps were implemented. The polymer solution was prepared using PLA (10 *w*/*v*) (ME33-FM-000150, Goodfellow, Pontevedra, Spain) dissolved in dichlomethane (DCM)/N,N-dimethylformamide (DMF) (70/30 *v*/*v*) (Scharlab, Barcelona, Spain) stirred continuously until a homogeneous solution was obtained. Next, this solution was placed in a syringe with a needle of 0.15 mm inner diameter connected to a high-voltage power supply. The electrospinning process was performed in a solution flow rate of 3 mL/h, at 20 cm needle-tip-to-collector distance and 20 kV of voltage. The fluid was ejected from the needle towards an aluminum sheet placed on the grounded collector of 15.77 cm diameter and 32 rps. Finally, the nanofibers were dried in a fixed vacuum system at room temperature for 48 h to remove solvent traces.

For the synthesis of BDMC-loaded PLA fibers, two contents of the drug were added to the PLA solution and stirred again, now in the dark because of the photosensitivity of curcumin. BDMC was supplied by Spectrum Chemical Mfg Corp for handling BDMC and was dissolved in DMSO (40 mg/mL). This concentration was used as a stock solution.

In order to determine the mass (m) of BDMC to be added to the PLA solution in the electrospinning device, a reference concentration (*c**) to be released in culture was taken from the literature [27], of *c** = 5 µM. The mass m was then calculated as
$$m=C_{PLA}\;V_{ES}\frac{c^{*}V}{t\;A\;d_{PLA}}$$
where *C_PLA_* is the concentration of PLA in the solution, *V_ES_* is the volume of the electrospun solution, *V* is the volume of culture medium, *d_PLA_* is the density of PLA, *t* is the disk thickness, and *A* is the disk area. Since we expect that only a part of the total BDMC-loaded in the disks will be released during the experiment, we decided to prepare samples with BDMC contents 5 times and 10 times the value determined as explained above. For our disks of *A* = 63.62 mm, *t* = 500 nm, and PM = 368.68 g/mol, this gave masses of BDMC per disk of m_d×5_ = 9.22 µg and m_d×10_ = 18.43 µg.

### 4.2. Morphology Characterization by FESEM

Samples were coated with platinum as an electrically conductive material. Field emission scanning electron microscopy (FESEM) images were obtained at 1 kV of acceleration voltage using a Zeiss Ultra 55 microscope (WITec GmbH, Barcelona, Spain). The images were further analyzed to examine fiber diameters and alignment. The average fiber diameters were obtained using the Image J software (National Institutes of Health, Bethesda, BD, USA).

### 4.3. Fourier-Transform Infrared Spectroscopy (FTIR)

FTIR analysis was performed to characterize functional groups in the nanofibers in order to detect any possible chemical interaction or modification between phases. The chemical characteristics of pure BDMC, PLA, and BDMC-loaded PLA nanofibers were determined using a Cary 630 FTIR (Agilent Technologies, Barcelona, Spain). The spectra were obtained in the ATR mode (Attenuated Total Reflection) from averages of 24 scans at 4 cm^−1^ from 4000 to 400 cm^−1^.

### 4.4. Differential Scanning Calorimetry (DSC)

DSC analysis of membranes was performed using a DSC 8000 (Perkin Elmer, Madrid, Spain) equipped with an Intracooler. Samples were exposed to a heating cycle in the range of 40 to 190 °C with a heating rate of 20 °C/min under the nitrogen atmosphere. An empty pan was used as a reference.

### 4.5. Thermogravimetric Analysis (TGA)

The thermogravimetric analysis of electrospun BDMC-loaded PLA nanofibers was performed using a TGA/SDTA 851 Mettler Toledo with STARe software. The materials of known weight were put in an aluminum pan and the analysis was carried out in an atmosphere of nitrogen. The heating ramp was applied from 30 °C to 800 °C with a heating rate of 10 °C/min. As a result, thermograms were obtained, which represent the mass loss as a function of temperature.

The results were calculated according to the following equation:$$X=\frac{R-P}{B-P}$$
where X represents the mass fraction, *R* is the percentage of weight loss of the sample, *P* is the percentage of weight loss of the PLA sample, *B* is the percentage of weight loss of the BDMC sample.

### 4.6. Loading Efficiency of BDMC

The drug content in each membrane was measured according to the procedure described below. For this purpose, the studies of BDMC loading were performed in DMSO. Aliquots of each membrane were measured by absorbance at fixed intervals of time (10, 20, 30, and 40 min) at the excitation of 426 nm using an ultraviolet-visible (UV–Vis) spectrometer. DMSO was replaced with fresh after each point. A calibration curve was established relating the weight of BDMC with absorbance to determine the amount of BDMC incorporated in the membrane. The absorbance values obtained from the same sample were added in order to calculate the total amount of BDMC loaded in the membrane. All measurements were performed in triplicate and the average values were taken.

### 4.7. In Vitro Drug Release Assay

In vitro drug release from membranes was evaluated by the dissolution technique. The in vitro BDMC release study was tested in a phosphate buffer saline containing 0.5% Tween-20 (PBST). Samples were immersed in PBST maintained at 37 °C. The amount of BDMC released at various times, from 1 h up to 60 days, was determined using an ultraviolet-visible (UV–Vis) spectrometer. A new PBST solution was employed for each measurement. The percentage of BDMC released versus time was calculated with the aid of a calibration curve of BDMC measured in the same conditions.

### 4.8. Schwann Cell Culture

SCs (P10301, Innoprot, Derio, Spain) were cultured on the e-spun membranes. First, cells were grown in F75 flasks to confluence replacing the culture medium every 2–3 days. The medium used was SCs medium kit (P60123, Innoprot, Spain) supplemented with growth factors. Cells were detached with 1% trypsin and counted in a Neubauer chamber.

Before seeding, materials were sanitized by UV in a laminar flow cabinet for 1 h on each side. They were preconditioned overnight in an SCs culture medium at 37 °C. SCs were seeded at a density of 10^5^ cells/sample suspended in 3 μL of medium and incubated for 1 h without medium to preserve cell–material adhesion. Next, the SCs culture medium used before with materials was added and the cells were cultured for different days at 37 °C and 5% CO_2_.

### 4.9. MTS Cell Proliferation Assay

The biocompatibility of fabricated electrospinning was assessed for culturing and proliferation of SCs by the colorimetric MTS assay (CellTiter 96 Aqueous One Solution Cell Proliferation Assay, Promega, Madrid, Spain). Cell proliferation was determined by a tetrazolium compound (3-(4,5-dimethylthiazol-2-yl)-5-(3-carboxymethoxyphenyl)-2-(4-sulfophenyl)-2H-tetrazolium, inner salt (MTS assay) after 1, 3, and 7 days of cell culture. The cells were washed with sterile PBS to avoid color interference from the compound and then incubated for 3 h with MTS reagent. Finally, the medium was removed and its absorbance was measured at 490 nm in a plate reader (Victor Multilabel Counter 1420 spectrophotometer, Perkin Elmer, Madrid, Spain).

### 4.10. Immunocytochemistry

After 24 h of cell culture, the samples were rinsed with PBS, fixed with 4% paraformaldehyde (PFA) (Panreac) for 20 min at room temperature, and washed twice with PBS 0.1 M. PLA/BDMC composite membranes autofluorescence, due to the presence of curcuminoid, was removed washing with decreased concentrations of DMSO until they lost their characteristic yellow color and rehydrated. Samples were blocked with 10% normal goat serum and cells were permeabilized with 0.1 vol.% Triton X-100 for 1 h at room temperature. After, the samples were incubated with rabbit polyclonal anti-S100β (1:200, ab52642, Abcam, Madrid, Spain) for 1 h at room temperature in the dark, washed twice with PBS, and were incubated with goat anti-rabbit Alexa 488 (A11008, Fisher Scientific, Madrid, Spain). Finally, nuclei were visualized with 4,6-diamidino-2-phenylindole dihydrochloride (DAPI; 1:5000, Fisher Scientific, Spain). For microscopy observation, the materials were mounted with Fluoroshield mounting medium (ab104139, Abcam).

### 4.11. Activation of NLRP3 Inflammasome

SCs were cultured at a concentration of 2 × 10^5^ cells/membrane. At 24 h, the percentage of FBS in culture media was decreased from 5% to 1%. After a minimum of 4 h, 10 µg/mL of LPS (ref L8274, Merck, Madrid, Spain) was added for priming. Twenty-four hours after, cells were stimulated with 5 mM ATP (ref A3127, Merck) 30 min before collection.

### 4.12. RNA Isolation and Real-Time Quantitative PCR (RT-qPCR) Analysis

For total RNA isolation, TriReagent (Merck) was used according to the manufacturer’s instructions. The amount of RNA was measured with a spectrophotometer. Additionally, 250 ng of RNA were reverse transcribed into cDNA using High-Capacity cDNA Reverse Transcription Kit (ThermoFisher Scientific, Madrid, Spain).

Primers were designed using UCSC and Primer3 and their sequences are presented in Table 1.

cDNA was amplified in a thermal cycler (LightCycler Instrument, Roche Diagnostics, Barcelona, Spain) using LightCycler 480 SYBR Green I Master (Roche, Barcelona, Spain). Amplification reactions were performed at 95 °C for 10 min, followed by 40 three-temperature cycles (20 s at 94 °C, 20 s at 60 °C, and 20 s at 72 °C). Once amplifications were finished, we performed a melting curve analysis in order to check the primers’ efficiency. The mRNA levels of GAPDH were selected as control and all samples were carried out in triplicate. The 2^−ΔΔCt^ method was used to calculate the relative gene expression of target genes.

### 4.13. Immunoblotting Assays

For protein isolation, cultured cells were lysed in RIPA lysis buffer with protease inhibitors for 30 min on ice. The concentration of protein was detected by Pierce BCA protein assay kit (ThermoFisher, Madrid, Spain). Protein was resolved on SDS-PAGE gel transferred onto PVDF membranes (IPVH00010, Millipore). The membranes were blocked with BSA 5% in Tween-Tris buffered saline before incubation overnight with specific antibodies: rabbit polyclonal anti-Caspase-1 (ab1872, Abcam, Madrid, Spain), mouse monoclonal anti-NLPR3 (ab205-0014, Adipogen, San Diego, CA, USA), IL-18 (sc-7954, Santa Cruz Biotechnology, Madrid, Spain), and mouse monoclonal anti-GAPDH (MAB374, Merck, Madrid, Spain). After, the membranes were incubated with their respective mouse and rabbit anti-HRP secondary antibodies (A9044 and A9169, Merck, Madrid, Spain) and a signal was developed by the ECL system (ECL Plus; Thermo Scientific, Madrid, Spain). Band density was analyzed with the Image J software (1.52p version).

### 4.14. Statistical Analysis

Three different samples of each material were studied. Statistical analyses were performed using a one-way or two-way analysis of variance (ANOVA) with *p* < 0.05 and Tuckey’s HSD test for a multiple-sample mean comparison. Results are expressed as mean ± standard deviation from at least three replicates. The significance between groups was studied using Graphpad software (Prism 9.0.2). Statistical significance is indicated by * *p* < 0.05, ** *p* < 0.01, *** *p* < 0.001, and **** *p* < 0.0001.

## 5. Conclusions

In the present study, we synthesized aligned PLA/BDMC composite membranes with the purpose to obtain an in situ gradual drug release system. We fabricated e-spun fibers to combine properties such as strength and lightness and a large surface area to volume ratio. BDMC, a curcuminoid with therapeutic properties, maintained its biomedical functional properties after an electrospinning process and could be released quickly after tissue damage when the inflammation response is activated by diffusion phenomenon. Additionally, our study provides evidence that our aligned fibers were effective in linearly guiding cell growth. Our findings revealed that BDMC in vitro promotes SCs proliferation and reduces inflammation by altering NLRP3 inflammasome activation and the subsequent release of the mature proinflammatory cytokines as IL-18. Both processes play essential roles in axon regeneration after PNI.

Taken together, these results show that PLA/BDMC composite membranes, in particular, the B1 membrane, could be employed in tissue engineering regeneration; however, further efforts are needed to explore this therapeutic strategy.

## Figures and Tables

**Figure 1 ijms-24-10340-f001:**
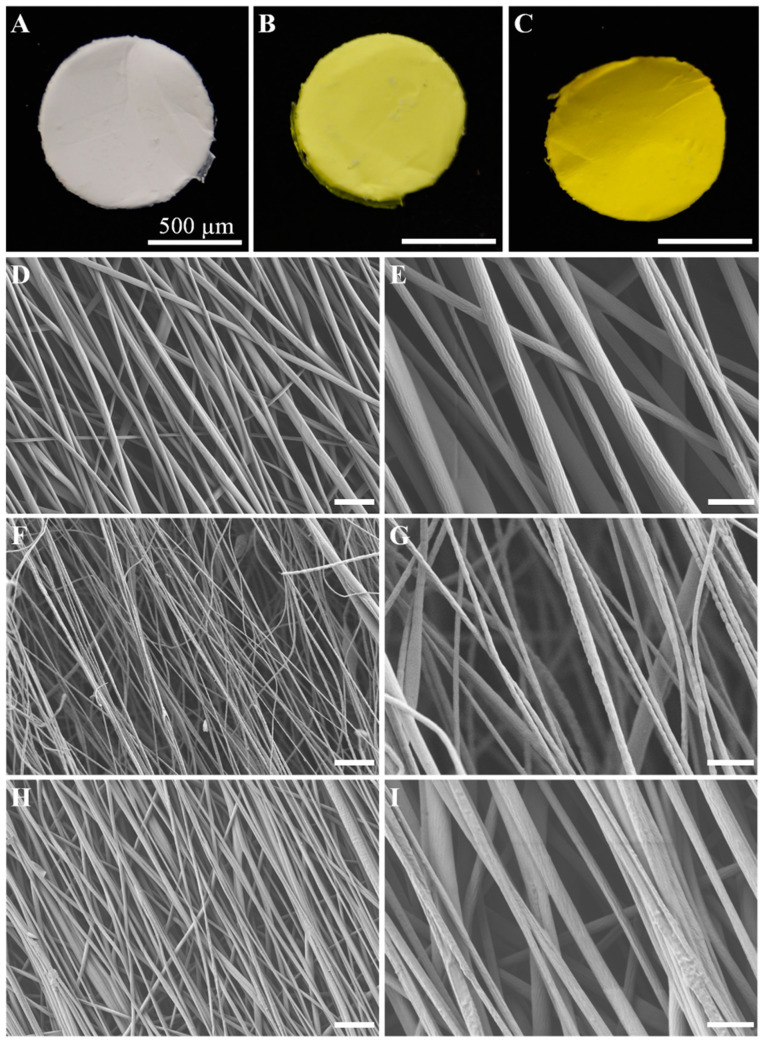
Macroscopic and FESEM images of the nanofiber membranes. (**A**) PLA e-spun membrane. (**B**,**C**) B1 and B2 e-spun membrane, respectively. (**D**,**E**) PLA nanofibers, (**F**,**G**) BDMC-loaded PLA nanofibers of B1 membrane, (**H**,**I**) BDMC-loaded PLA nanofibers of B2 membrane. Magnification 2000× and 7000×. Scale Bar: 2 µm.

**Figure 2 ijms-24-10340-f002:**
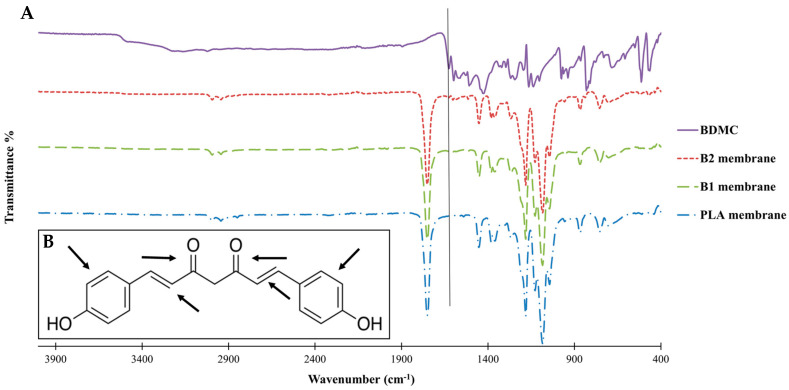
(**A**) FTIR spectrum of pure BDMC, BDMC-loaded PLA nanofibers of B1 and B2 membranes, and electrospinning pure PLA membrane. (**B**) Chemical structures of BDMC [1,7-bis(4-hydroxyphenyl)-1,6-heptadiene-3,5-dione].

**Figure 3 ijms-24-10340-f003:**
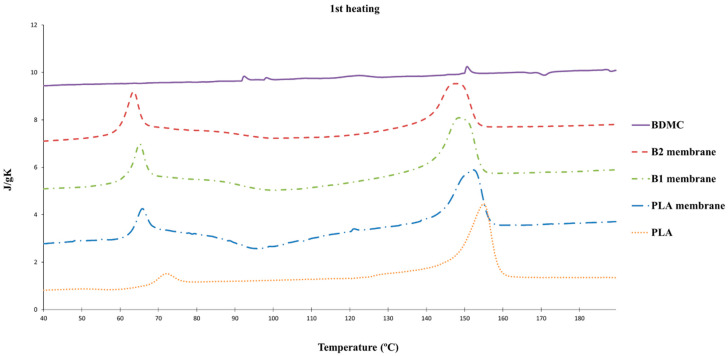
DSC curves of PLA, BDMC, and PLA e-spun fibers and PLA e-spun fibers loaded with different amounts of BDMC.

**Figure 4 ijms-24-10340-f004:**
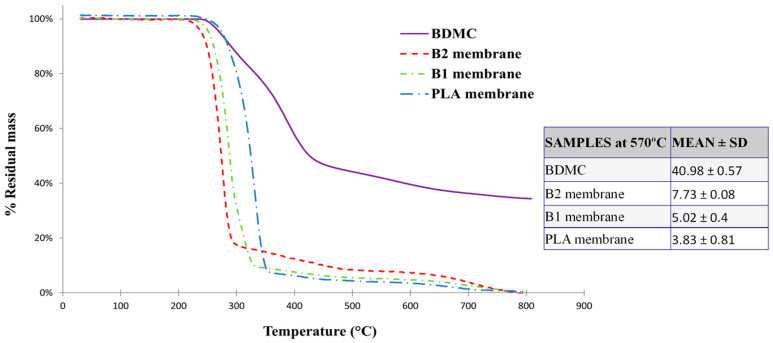
TGA thermograms of BDMC, PLA fibers, and BDMC-loaded PLA fibers.

**Figure 5 ijms-24-10340-f005:**
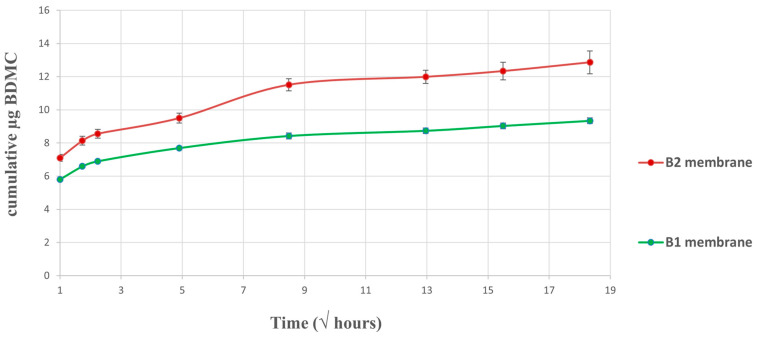
Drug release in vitro: shows the release profile of the drug at different time intervals in PBST buffer. The diffusion coefficient is obtained by the slope of the release curve.

**Figure 6 ijms-24-10340-f006:**
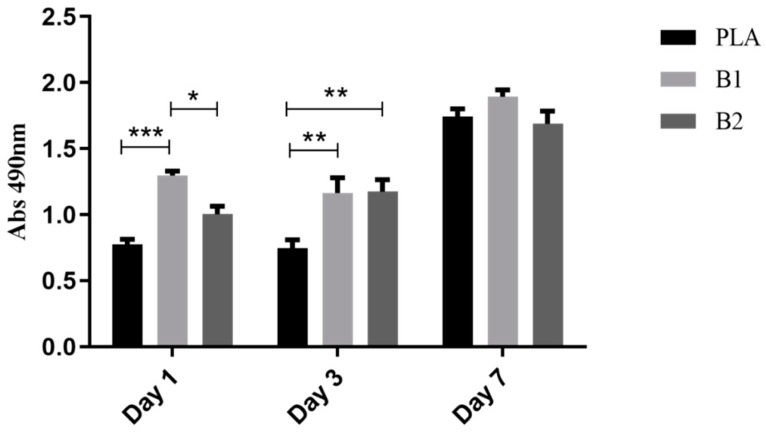
MTS assay data for SCs cultured on PLA, B1, and B2 membranes at 1, 3, and 7 days. Each column and bar represent the mean  ±  SEM (*n* = 4). Statistical significance is indicated by * *p* < 0.05, ** *p* < 0.01, and *** *p* < 0.001.

**Figure 7 ijms-24-10340-f007:**
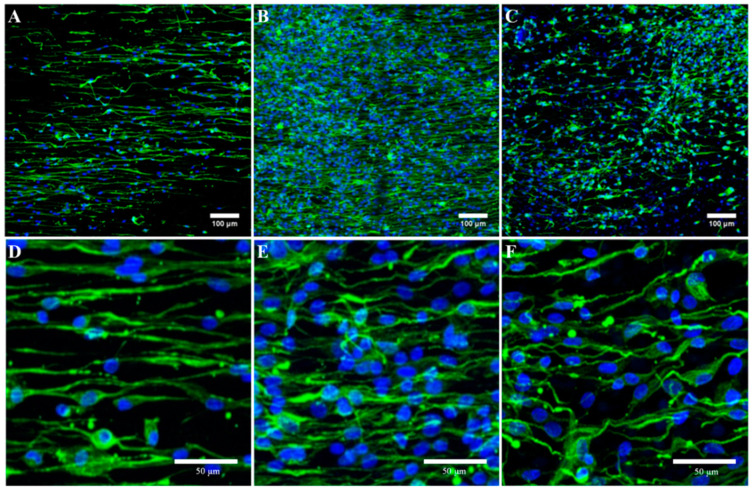
Immunofluorescence staining for S100β of SCs cultured for 1 day. (**A**,**D**) PLA membranes (**B**,**E**) B1 membranes and (**C**,**F**) B2 membranes.

**Figure 8 ijms-24-10340-f008:**
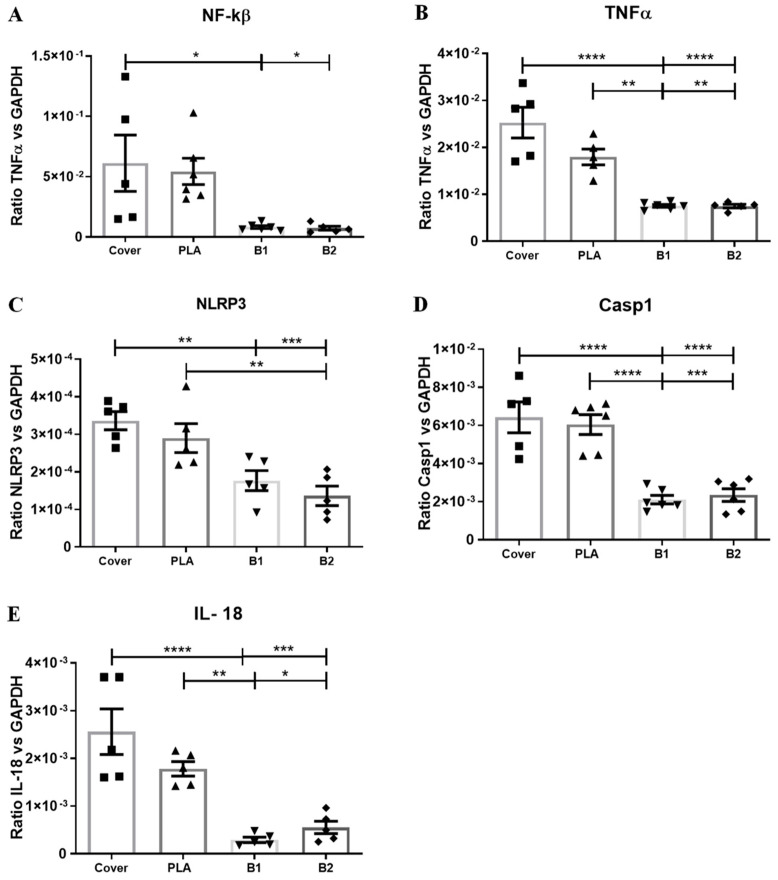
Statistical graphs of mRNA expressions measured with RT-qPCR analysis. (**A**,**B**) mRNA levels of the regulator of innate immunity NF-Kβ and the pro-inflammatory cytokineTNFα. (**C**,**D**) mRNA levels of inflammasome components NLRP3 and Casp1. (**E**) mRNA levels of proinflammatory cytokine IL-18. Each column and bar represent the mean  ±  SEM (*n* = 5). Statistical significance is indicated by * *p* < 0.05, ** *p* < 0.01, *** *p* < 0.001, and **** *p* < 0.0001.

**Figure 9 ijms-24-10340-f009:**
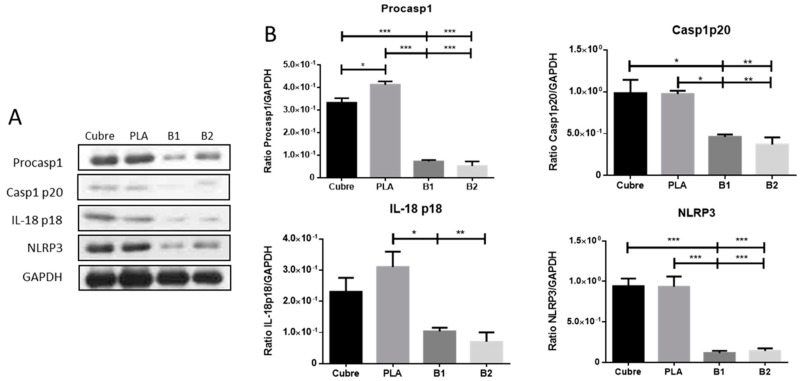
(**A**) Representative blots of Casp1, IL-18, and NLRP3. (**B**) Each column and bar represent the mean  ±  SEM *(n* = 3). Statistical significance is indicated by * *p* < 0.05, ** *p* < 0.01, and *** *p* < 0.001.

**Figure 10 ijms-24-10340-f010:**
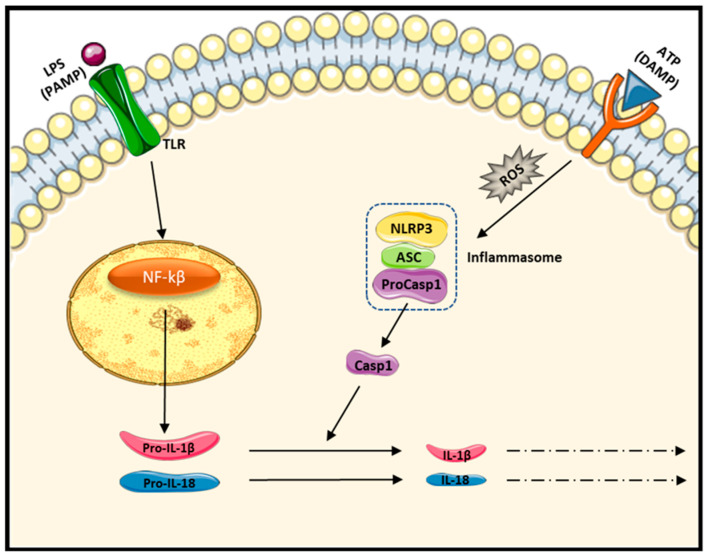
NLRP3 inflammasome priming and activation.

**Table 1 ijms-24-10340-t001:** List of analyzed genes and their respective sequences.

Gene	Primer Forward	Primer Reverse
NF-kβ	GGCAGAAGTCAACGCTCAG	GGTGTCGTCCCATCGTAGGT
TNF-α	TCATTCAAGGGCTGGTGAG	CGGCTTTGTGGAGGATTC
NLRP3	GAAGATTACCCACCCGAGAA	CCCAGCAAACCTATCCACTC
Casp-1	GCAAGCCAGATGTTTATCACTT	CGCCACCTTCTTTGTTCACA
IL-18	ACAGCCAACGAATCCCAGAC	GACATCCTTCCATCCTTCACA
GAPDH	AGACAGCCGCATCTTCTTGT	CTTGCCGTGGGTAGAGTCAT

## Data Availability

All the data generated in this research are included in the manuscript.

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
