# Peer review of "Characterization of Electrospun BDMC-Loaded PLA Nanofibers with Drug Delivery Function and Anti-Inflammatory Activity"

_ijms, 2023, doi:10.3390/ijms241210340_

Round 1

Reviewer 1 Report

Report on “Characterization of electrospun BDMC loaded PLA nanofibers with drug delivery function and anti-inflammatory activity”.

This manuscript entitled "Characterization of electrospun BDMC loaded PLA nanofibers with drug delivery function and anti-inflammatory activity" reports the development of BDMC loaded nanofibers and their in vitro biological evaluation in terms of stimulating Schwann cell growth.

General comment:

The manuscript contains some confusing sentences, such as: "...electrospinning is a versatile and cost-effective method for biomedical applications and, are promising as drug carrier candidates due to their properties to mimic the extracellular matrix", which inhibits the reading flow. Part of the discussion on the differences between formulations also does not consider the addition of DMSO to the formulation, although it is possible that it plays an important role in the morphological and consequently biological properties of the nanofibers produced. In addition, the authors claim that electrospinning is a suitable method for the preparation of biomedical materials. Do the authors agree that the use of harmful solvents such as DMF, DCM and DMSO (line 111) is suitable for the industrial production of drugs? In addition, the production rate used here was 3 ml/hour (line 115). Is this rate compatible with an industrial scale?

Please find below the detailed comments:

·      Abstract, line 1 (also appears later) ïƒ  The authors interchange the use of the words "delivery" and "release", e.g. the term "controlled drug release systems" is different from "controlled drug delivery systems", please check which term fits better in the paper and unify it throughout the text.

·      Abstract, line 20 ïƒ  It seems that electrospinning is the “thing” that can mimic the extracellular matrix, not nanofibers, that sounds strange.

·      Abstract, line 31 ïƒ  The authors conclude the abstract with a speculation about the use of their nanofibers, however, no living cells or complex structures were studied here. Also, this conclusion is not consistent with the Conclusion section of the manuscript.

·      Introduction, line 79 ïƒ  Please describe the abbreviation “PNI”.

·      Introduction, line 103 ïƒ  Please rephrase the sentence to clarify that the work is based on the evaluation of two concentrations of BDMC.

·      Introduction, line 105 ïƒ  As in the abstract, the authors speculate too much about the possibilities of their work, and they are not mentioned in the sections where they could be brought in more, such as discussion and conclusion.

·      Methods, line 112 ïƒ  Please indicate if the ratio is m/m, m/v, or v/v for all concentrations in percent in the text.

·      Methods, line 118 ïƒ  Please better describe the vacuum system used to evaporate the remaining solvent.

·      Methods, line 126 ïƒ  The "C" in equation 1 does not match the "C" in the text.

·      Methods, line 127 ïƒ  The correct abbreviation for molecular mass is "MW", please correct it.

·      Methods, line 128 ïƒ  what do the authors mean by "weigh of BDMC bar"?

·      Methods, line 153 ïƒ  what do the authors mean by "cup"?

·      Methods, line 159 ïƒ  why the conditions used for TGA are not comparable to those used for DSC even though the targets were different, the heating rate may have an effect on the behavior of the sample.

·      Methods, line 167 ïƒ  Confusing sentence (grammar), please revise.

·      Methods, line 167 ïƒ  Please revise the use of the word "release", it seems that "load" is a more appropriate word.

·      Methods, line 178 ïƒ  How did the authors achieve this Tween concentration. It could also be better described why the tween is needed.

·      Methods, line 178 ïƒ  Remove the parenthesis at PBST if it is cited later.

·      Methods, line 183 ïƒ  Why was no control performed without a membrane? How do we know if membranes are better than standard conditions?

·      Methods, line 187 ïƒ  The Neubauer chamber was used for counting. So how did the authors control for cell viability?

·      Methods, line 190 ïƒ  It seems that the number of cells per sample is wrong, 105 instead of 1x105, please check.

·      Methods, line 194 ïƒ  The MTS assay gives no indication of cell proliferation as it is a method that only assesses some enzymatic structures of mitochondria. Please review the methodology and discussion on this technique.

·      Methods, line 218 ïƒ  Please provide the initial FBS concentration that was likely used in other tests.

·      Methods, line 241 ïƒ  What is the meaning of “TBST buffer” expression?

·      Results, line 260 ïƒ  What do the authors mean by "beads" and why did they expect them?

·      Results, line 268 ïƒ  Please indicate which diameter corresponds to which formulation (B1 and B2). Also, the authors incorrectly state that the addition of BDMC narrows the fibers, which does not seem to be the case when calculating the relative standard deviation, please clarify.

·      Results, line 272 ïƒ  As mentioned by this reviewer, the authors should further investigate the effect of DMSO on the morphological properties of the nanofibers. True control nanofibers would indeed have been produced by using the same amounts of DMSO that were used to produce B1 and B2.

·      Results, line 274 ïƒ  Figure 1, was FESEM or SEM (see Methods section) the technique used?

·      Results, line 278 ïƒ  The reviewer suggests that all images should have scale bars.

·      Results, line 293 ïƒ  Figure 2, the reviewer suggests identifying the part of the molecule of interest.

·      Results, line 309 ïƒ  Is it the addition of BDMC or DMSO that causes this difference in the thermal properties of the nanofibers?

·      Results, line 236 ïƒ  Similar to the morphological and DSC characterization, the effect of DMSO on the formulations should be further investigated.

·      Results, line 330 ïƒ  Please better describe the following statement/hypothesis: "Loss of thermal stability of PLA/BDMC composite membranes may be attributed to the smaller diameter of the fibers."

·      Results, line 341 ïƒ  The correct term is "in vitro drug release assay"

·      Results, line 342 ïƒ  Even though this is visible in Figure 6, authors should provide "rate values" when describing kinetics.

·      Results, line 343 ïƒ  Can these nanofibers be called "Controlled drug delivery systems" if their release is so rapid? Please unify the figure and text, or % or microgram units.

·      Results, line 352 ïƒ  The text is unrelated to the manuscript.

·      Results, line 367 ïƒ  Neither here nor later do the authors hypothesize better performance of the B1 formulation, which should be clearly stated to increase interest in studying this system/molecule.

·      Results, lines 400 and 403 ïƒ  what do the authors mean by "very significant" and "highly significant"?

·      Discussion, line 438 ïƒ  It is stated that BDMC is soluble in DCM and DFA. Why then was DMSO used?

·      Discussion, line 438 ïƒ  What PLA was used? Was it a pure PLLA or a mixture?

·      Discussion, line 442 ïƒ  Please provide more information on this topic, at least the standard deviation values.

·      Discussion, line 457 ïƒ  Again, the authors neglect the effect of adding DMSO to the formulation. Please revise the whole discussion.

·      Discussion, line 477 ïƒ  What is the basis for the following statement "... most of the BDMC was loaded at the surface of the fibers due to the decrease in the average diameter of the fibers..." ??

·      Discussion, line 492 ïƒ  What do the authors mean by "SCs maduration"?

·      References ïƒ  Only 7 of 50 references are from the last 5 years.

English should be revised as indicated in the "Comments and Suggestions" section.

Author Response

                                                                                       Valencia, 1st June 2023

Reviewer 1

This manuscript entitled "Characterization of electrospun BDMC loaded PLA nanofibers with drug delivery function and anti-inflammatory activity" reports the development of BDMC loaded nanofibers and their in vitro biological evaluation in terms of stimulating Schwann cell growth.

General comment:

The manuscript contains some confusing sentences, such as: "...electrospinning is a versatile and cost-effective method for biomedical applications and, are promising as drug carrier candidates due to their properties to mimic the extracellular matrix", which inhibits the reading flow. Part of the discussion on the differences between formulations also does not consider the addition of DMSO to the formulation, although it is possible that it plays an important role in the morphological and consequently biological properties of the nanofibers produced. In addition, the authors claim that electrospinning is a suitable method for the preparation of biomedical materials. Do the authors agree that the use of harmful solvents such as DMF, DCM and DMSO (line 111) is suitable for the industrial production of drugs? In addition, the production rate used here was 3 ml/hour (line 115). Is this rate compatible with an industrial scale?

Thank you very much for your question. We are aware that these solvents have toxic effects and for this reason the fibers were subjected to a process (vacuum system for 48h) to eliminate traces of solvent. Subsequently, it was shown with TGA that there were no traces.

We appreciate your comment. We have not considered the industrial scaling of this procedure. The flow used corresponds to a laboratory equipment and was obtained by optimizing the different parameters of the equipment.

Please find below the detailed comments:

  • Abstract, line 1 (also appears later). The authors interchange the use of the words "delivery" and "release", e.g. the term "controlled drug release systems" is different from "controlled drug delivery systems", please check which term fits better in the paper and unify it throughout the text.

The term "release" has been unified throughout the manuscript.

  • Abstract, line 20. It seems that electrospinning is the “thing” that can mimic the extracellular matrix, not nanofibers, that sounds strange.

Thank you very much. We have corrected the unclear text. 

Abstract, line 31. The authors conclude the abstract with a speculation about the use of their nanofibers, however, no living cells or complex structures were studied here. Also, this conclusion is not consistent with the Conclusion section of the manuscript.

Thank you very much for your comment. We have modified the different sections as suggested by the reviewer.

  • Introduction, line 79. Please describe the abbreviation “PNI”.

Thank you very much. We have described the abbreviation.

PNI: Peripheral nerve injury

  • Introduction, line 103. Please rephrase the sentence to clarify that the work is based on the evaluation of two concentrations of BDMC.

Modified as requested. We have corrected the unclear sentence:

In this work, we prepared PLA/BDMC composite membranes by electrospinning at two different BDMC concentrations and evaluate their physical properties and in vitro behaviour.

  • Introduction, line 105As in the abstract, the authors speculate too much about the possibilities of their work, and they are not mentioned in the sections where they could be brought in more, such as discussion and conclusion.

Thank you very much for your appropriate comment. We have changed the final sentences of the abstract and discussion with more modest ones on the potential use of PLA/BDMC composite membranes.

  • Methods, line 112. Please indicate if the ratio is m/m, m/v, or v/v for all concentrations in percent in the text.

Thank you, they have been corrected. This sentence has been introduced in the new version of the manuscript:

“The polymer solution was prepared using PLA (10 w/v) (ME33-FM-000150, Goodfellow) dissolved in dichlomethane (DCM) / N,N-dimethylformamide (DMF) (70/30 v/v) (Scharlab, Spain) stirred continuously till a homogeneous solution was obtained.”

  • Methods, line 118. Please better describe the vacuum system used to evaporate the remaining solvent.

We appreciate your comment and have included a better description of the mechanism used to remove solvent residues.

  • Methods, line 126. The "C" in equation 1 does not match the "C" in the text.

We thank the reviewer for drawing our attention. The calculations have been redone to try to make them more understandable.

  • Methods, line 127. The correct abbreviation for molecular mass is "MW", please correct it.

Thank you. It has been corrected

  • Methods, line 128. what do the authors mean by "weigh of BDMC bar"?

The calculations have been redone to try to make them more understandable.

  • Methods, line 153. what do the authors mean by "cup"?

Thank you very much for drawing our attention. The word cup has been modified for pan.

  • Methods, line 159. why the conditions used for TGA are not comparable to those used for DSC even though the targets were different, the heating rate may have an effect on the behavior of the sample.

Thank you very much for you comment. We have used in the DSC and TGA standard heating rates. Both do not have to coincide because in DSC the important phenomenon is a heat transfer in the sample that is in a closed capsule. In TGA, however, the sample is in an open capsule and the relevant phenomenon is the diffusion of the volatiles resulting from thermodegradation.

  • Methods, line 167. Confusing sentence (grammar), please revise.

Thank you very much for drawing our attention, and we apologize for these mistakes. The new sentence is: “Aliquots of each membrane were measured by absorbance at fixed intervals of time (10, 20, 30 and 40 min) at the excitation of 426 nm using an ultraviolet-visible (UV–Vis) spectrometer.”

  • Methods, line 167. Please revise the use of the word "release", it seems that "load" is a more appropriate word.

Thank you. The word has been changed as suggested.

  • Methods, line 178. How did the authors achieve this Tween concentration. It could also be better described why the tween is needed.

Thank you very much for your comment. Tween is a non-ionic surfactant commonly used in drug delivery applications for dispersing of hydrophobic drugs.

Here are examples of other works using Tween to study in vitro curcumin release.

Bibliography: 2019Cellulose 26(4) DOI: 10.1007/s10570-019-02445-6

“Curcumin/Tween 20-incorporated cellulose nanoparticles with enhanced curcumin solubility for nano-drug delivery: characterization and in vitro evaluation.”

Molecules 2022 Mar 8;27(6):1759. doi: 10.3390/molecules27061759.

“A New and Sensitive HPLC-UV Method for Rapid and Simultaneous Quantification of Curcumin and D-Panthenol: Application to In Vitro Release Studies of Wound Dressings”.

  • Methods, line 178. Remove the parenthesis at PBST if it is cited later.

Thank you. The parenthesis has been removed.

  • Methods, line 183. Why was no control performed without a membrane? How do we know if membranes are better than standard conditions?

Thank you very much for your question. Since we were interested in testing the effectiveness of the electrospinning membrane as a BDMC release vehicle, our control consisted of a PLA membrane without BDMC.

  • Methods, line 187. The Neubauer chamber was used for counting. So how did the authors control for cell viability?

Thank you very much for your question. We counted the cells with the neubauer chamber in order to seed the same concentration of cells on all the membranes. Later, an MTS Assay was performed at different times to study cell viability and citotoxicity at both concentrations of BDMC using as control the PLA membrane.

  • Methods, line 190.It seems that the number of cells per sample is wrong, 105 instead of 1x105, please check.

Thank you very much for drawing our attention, and we apologize for this mistake.

  • Methods, line 194. The MTS assay gives no indication of cell proliferation as it is a method that only assesses some enzymatic structures of mitochondria. Please review the methodology and discussion on this technique.

Thank you very much for you comment. The CELLTITER 96® AQUEOUS ONE SOLUTION CELL PROLIFERATION ASSAY used in this work is a colorimetric method for determining the number of viable cells in proliferation, cytotoxicity or chemosensitivity assays.

Bibliography: Assay Guidance Manual [Internet]. Bethesda (MD): Eli Lilly & Company and the National Center for Advancing Translational Sciences; 2004.

2013 May 1 [updated 2016 Jul 1].

PMID: 23805433 Bookshelf ID: NBK144065

“Cell Viability Assays”

  • Methods, line 218. Please provide the initial FBS concentration that was likely used in other tests.

As suggested, we have included the initial FBS concentration, 5%. This percentage has been calculated from the commercial kit provided by Innoprot.

  • Methods, line 241. What is the meaning of “TBST buffer” expression?

Thank you, it has been corrected. It was referring to tween-tris buffered saline.

  • Results, line 260. What do the authors mean by "beads" and why did they expect them?

Beads are protrusions that appear randomly on the nanofibers when the parameters of the electrospinning process are not well adjusted. The absence of beads on the fiber surface indicates a good efficiency of the electrospinning process.

We expected beadless nanofibers because previously our electrospinning parameters had been optimized to avoid beads formation.

Bibliography: April 2008 Polymer International 57(4):632-636 DOI: 10.1002/pi.2387

“Controlling numbers and sizes of beads in electrospun nanofibers.”

PMID: 35228788 PMCID: PMC8867693 DOI: 10.1007/s11837-022-05180-9. Epub 2022 Feb 24.

“A Review on Curcumin-Loaded Electrospun Nanofibers and their Application in Modern Medicine”

  • Results, line 268. Please indicate which diameter corresponds to which formulation (B1 and B2). Also, the authors incorrectly state that the addition of BDMC narrows the fibers, which does not seem to be the case when calculating the relative standard deviation, please clarify.

Thank you very much for your comment. We have corrected the unclear text.

There is a significant difference between the diameter of PLA fibers and PLA/BDMC fibers. This difference is not directly proportional to the amount of drug loaded, possibly due to other factors such as viscosity and conductivity of the solution as discussed later in the manuscript.

  • Results, line 272. As mentioned by this reviewer, the authors should further investigate the effect of DMSO on the morphological properties of the nanofibers. True control nanofibers would indeed have been produced by using the same amounts of DMSO that were used to produce B1 and B2.

DMSO has a reportedly low toxicity and was only used to dissolve the BDMC. We made sure to remove any possible trace of the solvents with the vacuum system before using the membranes.

Bibliography: Nanomaterials 2019, 9(1), 52; https://doi.org/10.3390/nano9010052

 “New polymers for needleless electrospinning from low-Toxic Solvents”

  • Results, line 274. Figure 1, was FESEM or SEM (see Methods section) the technique used?

Thank you very much for your useful comment. We have now modified the text according to your suggestions. We used a FESEM microscopy.

  • Results, line 278. The reviewer suggests that all images should have scale bars.

We appreciate your suggestion and have included all the scale bars.

  • Results, line 293. Figure 2, the reviewer suggests identifying the part of the molecule of interest.

Thank you very much for your comment. Now figure 2 has been modified following the advice of the reviewer.

  • Results, line 309. Is it the addition of BDMC or DMSO that causes this difference in the thermal properties of the nanofibers?

Thank you very much for you comment. The differences in the thermal properties of the nanofibers is due to the BDMC, since the TGA shows that there are no residues of DMSO

  • Results, line 236. Similar to the morphological and DSC characterization, the effect of DMSO on the formulations should be further investigated.

Throughout all the work, we assume that the TGA results are conclusive to affirm that there are no DMSO residues (Figure 4).

  • Results, line 330. Please better describe the following statement/hypothesis: "Loss of thermal stability of PLA/BDMC composite membranes may be attributed to the smaller diameter of the fibers."

Now the sentence has been modified by this "the weight loss occurs in the samples with BDMC at lower temperatures. This may be due to the fact that in these samples the diameter of the nanofibers is smaller and therefore, the diffusion of volatile products of thermal degradation is easier”

  • Results, line 341. The correct term is "in vitro drug release assay"

Modified as requested by the reviewer.

  • Results, line 342.Even though this is visible in Figure 6, authors should provide "rate values" when describing kinetics.

We appreciate the suggestion. We have changed the figure and calculated the slope of the release curves to show the diffusion coefficient of BDMC in B1 and B2 membranes.

  • Results, line 343. Can these nanofibers be called "Controlled drug delivery systems" if their release is so rapid? Please unify the figure and text, or % or microgram units.

We have changed the expression to: “gradual release system because, as the reviewer mentions, we have no control over the release rate of the BDMC.”

  • Results, line 352. The text is unrelated to the manuscript.

The sentence has been removed from manuscript. Thank you for drawing our attention to this.

  • Results, line 367. Neither here nor later do the authors hypothesize better performance of the B1 formulation, which should be clearly stated to increase interest in studying this system/molecule.

Thank you very much for your comment. We have added a comment in the discussion section suggesting that since both BDMC membranes have similar anti-inflammatory effects, it might be more interesting for further work to use the B1 membrane. This membrane with half the amount of drug compared to B2, achieves a higher increase in cell proliferation in the short term. In addition, the manufacturing cost would be lower by using less amount of BDMC.

  • Results, lines 400 and 403. what do the authors mean by "very significant" and "highly significant"?

Thank you very much for you comment. These sentences have been revised and appropriately changed. Results are very or highly significant when compare to cells seeded on cover. When compared to the membrane controls the results are significant.

  • Discussion, line 438. It is stated that BDMC is soluble in DCM and DFA. Why then was DMSO used?

We dissolved the BDMC in DMSO because the other solvents mentioned are more volatile and this fact made it difficult to control the amount of BDMC effectively added.

  • Discussion, line 438.What PLA was used? Was it a pure PLLA or a mixture?

PLA used was: goodfellow me33-fm-000150. It was pure PLLA.

  • Discussion, line 442. Please provide more information on this topic, at least the standard deviation values.

As suggested, we have included the standard deviation values associated with the conductivity values.

  • Discussion, line 457. Again, the authors neglect the effect of adding DMSO to the formulation. Please revise the whole discussion.

Throughout all the work, we assume that the TGA results are conclusive to affirm that there are no DMSO residues (Figure 4). However, there is literature reporting that the addition of DMSO to the electrospinning solution produces an effect on the fibers diameter. We have commented it in the discussion of the article.

Bibliography: Polymer Journal, Vol. 39, No. 6, pp. 622–631 (2007)

#2007 The Society of Polymer Science, Japan

“Electrospun Gelatin Fibers: Effect of Solvent System on Morphology and Fiber Diameters.”

  • Discussion, line 477. What is the basis for the following statement "... most of the BDMC was loaded at the surface of the fibers due to the decrease in the average diameter of the fibers..." ??

We appreciate your comment and have changed the statement to this one: “This behavior indicated that most of BDMC was loaded on the surface of the fibers.

  • Discussion, line 492.What do the authors mean by "SCs maduration"?

Maduration is the process that allows a cell to become functional and specialized. the developmental stages of cells.

For example: SC development starts from neural crest cells, through SC precursors, immature SCs, pro-myelinating SCs to mature myelinating and/or non-myelinating SCs.

Bibliografía “Schwann cell development, maturation and regeneration: a focus on classic and emerging intracellular signaling pathways” Neural Regen Res. 2017 Jul; 12(7): 1013–1023.

doi: 10.4103/1673-5374.211172

But in this work we do not study this process, it is only mentioned because other works have demonstrated that aligned e-spun fibers promote SCs maduration.

  • References . Only 7 of 50 references are from the last 5 years.

Recent new references have been included in the manuscript.

Yours sincerely,

Cristina Martínez Ramos

Centre for Biomaterials and Tissue Engineering. Universidad Politécnica de Valencia

46022 Valencia, Spain

Email: cris_mr_1980@hotmail.com

Reviewer 2 Report

This manuscript deals with the characterization of electrospun BDMC loaded PLA nanofibers with drug delivery function and anti-inflammatory activity. The characterization was performed with SEM, FT-IR, DSC, and TGA. Furthermore, the loading efficiency, the drug release, the cell viability, the immunological response, and the inflammatory response of the nanofibers were analyzed. The manuscript is overall interesting, and the data support the suggestion in the conclusions. However, it is hard to recommend publication before the authors address the following points:

1.      The titles have serious problems. “3. Results and Discussions” and “4. Discussion” have been found. Is there any reason to repeat the discussion separately? The other problem is the titles of “2. Materials and Methods”. What is the criterium to suggest the titles? Some titles are a subject such as “Morphological characterization”, and the others are a method like “FT-IR”. Either one could be used as the criterium of a title, but consistency should be kept.

2.      It is unclear how the amount of BDMC was calculated. What does BDMC bar stand for? In order to estimate the amount, why m(BDMC) should be multiplied by m(PLA)? Is M “Weight” of BDMC bar, not “Weigh” of BDMC bar?

3.      In line 131, there is a description that the concentration increase was decided 5 and 10 times. If then, how to estimate or measure the degree of the increase? Please elaborate on the way of the estimation or the measurement.

4.      In line 268 to 269, there is a description that the BDMC addition narrowed the diameter distribution. However, the interpretation of the reason has been missing.

5.      Most of the figure fonts look too small to be recognized.

6.      Although Figure 6 shows that the cumulative amount of B2 BDMC is clearly much more than that of B1 BDMC, why their mRNA levels of Figure 9 are similar only except for NLRP3?

7.      (A-B, (C-D), and (E) have been described in Figure 9 caption, but they are missing in Figure 9.

Moderate editing would be fine.

Author Response

                                                                                           Valencia, 1st June 2023

Reviewer 2

This manuscript deals with the characterization of electrospun BDMC loaded PLA nanofibers with drug delivery function and anti-inflammatory activity. The characterization was performed with SEM, FT-IR, DSC, and TGA. Furthermore, the loading efficiency, the drug release, the cell viability, the immunological response, and the inflammatory response of the nanofibers were analyzed. The manuscript is overall interesting, and the data support the suggestion in the conclusions. However, it is hard to recommend publication before the authors address the following points:

  1. The titles have serious problems. “3. Results and Discussions” and “4. Discussion” have been found. Is there any reason to repeat the discussion separately? The other problem is the titles of “2. Materials and Methods”. What is the criterium to suggest the titles? Some titles are a subject such as “Morphological characterization”, and the others are a method like “FT-IR”. Either one could be used as the criterium of a title, but consistency should be kept.

We thank the reviewer for drawing our attention and have corrected the titles. First, in results section we comment about the results obtained in this work and, in the discussion section, we proceed to discuss the possible causes and others.

We have also proceeded to change other headings in the material and methods section to clarify them.

  1. It is unclear how the amount of BDMC was calculated. What does BDMC bar stand for? In order to estimate the amount, why m(BDMC) should be multiplied by m(PLA)? Is M “Weight” of BDMC bar, not “Weigh” of BDMC bar?

Thank you very much for your comment. We have done the calculations relating to how to determine the mass of BDMC to be added to a volume of electrospinning solution to obtain a target release.

  1. In line 131, there is a description that the concentration increase was decided 5 and 10 times. If then, how to estimate or measure the degree of the increase? Please elaborate on the way of the estimation or the measurement.

In order to determine the mass (m) of BDMC to be added to the PLA solution in the electrospinning device, a reference concentration (c*) to be released in culture was taken from the literature (Requejo-Aguilar R, Alastrue-Agudo A, Cases-Villar M, Lopez-Mocholi E, England R, Vicent MJ, Moreno-Manzano V. Combined polymer-curcumin conjugate and ependymal progenitor/stem cell treatment enhances spinal cord injury functional recovery. Biomaterials. 2017 Jan; 113:18-30. doi: 10.1016/j.biomaterials.2016.10.032), of c* = 5 µM. The mass m was then calculated as

m =

where CPLA is the concentration of PLA in the solution, VES is the volume of the electrospun solution, V is the volume of culture medium, dPLA is the density of PLA, t is the disk thickness, and A is the disk area. Since we expect that only a part of the total BDMC loaded in the disks will be released during the experiment, we decided to prepare samples with BDMC contents 5 times and 10 times the value determined as explained above. For our disks of A = 63.62mm, t = 500 nm, and PM = 368.68 g/mol this gave masses of BDMC per disk of md×5 = 9.22µg and md×10 = 18.43µg.

  1. In line 268 to 269, there is a description that the BDMC addition narrowed the diameter distribution. However, the interpretation of the reason has been missing.

Thank you very much for your comment. Lines 268 and 269 belong to the results section. The possible explanation of this decrease in the diameter of the fibers is made in the discussion section.

  1. Most of the figure fonts look too small to be recognized.

We appreciate your comment and have used a larger font size.

  1. Although Figure 6 shows that the cumulative amount of B2 BDMC is clearly much more than that of B1 BDMC, why their mRNA levels of Figure 9 are similar only except for NLRP3?

Thank you very much for your question. Including NLRP3, all messenger values are similar for both BDMC concentrations. It means that both doses have anti-inflammatory effect, but even if we increase the concentration of the drug the effect is not further potentiated. It is not dose-dependent.

  1. (A-B, (C-D), and (E) have been described in Figure 9 caption, but they are missing in Figure 9.

Thank you very much for drawing our attention, and we apologize for these mistakes. We have included the figure labels mentioned in the manuscript A, B, C, D and E.

WE WOULD LIKE TO THANK THE REVIEWER FOR HIS/HER HELP AND SUGGESTIONS TO CLARIFY SOME PARTS OF THE TEXT. ALL THE CORRECTIONS HAVE BEEN MADE.

Please find enclosed the new electronic version of the paper entitled “Characterization of electrospun BDMC loaded PLA nanofibers with drug delivery function and anti-inflammatory activity” by Maria José Morillo-Bargues, Andrea Olivos Osorno, Consuelo Guerri, Manuel Monleón Pradas and Cristina Martínez-Ramos, submitted to be considered for publication in International Journal of Molecular Sciences.

Yours sincerely,

Cristina Martínez Ramos

Centre for Biomaterials and Tissue Engineering. Universidad Politécnica de Valencia

46022 Valencia, Spain

Email: cris_mr_1980@hotmail.com

Reviewer 3 Report

My comments are in attachment.

Text quality should be improved, and figure mixture should be excluded.

Author Response

                                                                                         Valencia, 1st June 2023

Reviewer 3

This manuscript contains a description of the conditions used for the preparation of electrospun BDMC loaded PLA nanofibers. Then, the PLA/BDMC composite membranes were fabricated to reach a in situ-controlled drug delivery. BDMC is a curcuminoid with therapeutic properties. Thermal stability, morphology and drug release properties of the membranes were determined by conventional experimental techniques. The experimental conditions used for the sample preparation and property measurements are given in detail that opens a possibility for future testing of the results by other researchers. In my opinion, this study is valuable because new system for the drug release was successfully synthesized and its basic microstructural and thermal characteristics were determined. The general level of this study is good and manuscript could be considered for publication after minor revision reasonable to increase the text quality. My several corrections proposed for text are listed below for author consideration.

Page 1

Controlled drug release systems have been the subject of many investigations to achieve the therapeutic effect of drugs.

Controlled drug release systems are the subjects of many investigations to achieve the therapeutic effect of drugs.

AMENDED AS SUGGESTED BY THE REVIEWER

Page 1

They have numerous advantages such as localized effect, lower side effects and less onset of action.

They have numerous advantages, such as localized effect, lower side effects and less onset of action.

COMMA ADDED AS REQUESTED

Page 1

Among drug-delivery systems, electrospinning is a versatile and cost-effective method for biomedical applications and, are promising as drug carrier candidates due to their properties to mimic the extracellular matrix.

This sentence is unclear.

WE HAVE CORRECTED THE UNCLEAR TEXT

Page 1

In this work, electrospun fibers are made of Poly-L-lactic acid (PLA), one of the most widely tested material that has excellent biocompatible and biodegradable properties. A curcuminoid, bisdemethoxycurcumin (BDMC) is added in order to complete the drug delivery system. PLA/BDMC membranes were characterized, and biological characteristics were examinated in vitro.

In this work, electrospun fibers were made of Poly-L-lactic acid (PLA), one of the most widely tested material that has excellent biocompatible and biodegradable properties. A curcuminoid, bisdemethoxycurcumin (BDMC) was added in order to complete the drug delivery system. The PLA/BDMC membranes were characterized, and biological characteristics were examined in vitro.

MODIFIED AS SUGGESTED BY THE REVIEWER

Page 1

Results show that the average fiber diameters was reduced with the drug which was mainly released within the first 24 hours by diffusion mechanism. We have seen that the use of our membranes loaded with BDMC enhances the rate of proliferation in Schwann cells, the main peripheral neuroglial cells, and modulates inflammation by reducing NLRP3 inflammasome activation.

The results show that the average fiber diameter was reduced with the drug which was mainly released during the first 24 h by diffusion mechanism. It was seen that the use of our membranes loaded with BDMC enhanced the rate of proliferation in Schwann cells, the main peripheral neuroglial cells, and modulated inflammation by reducing NLRP3 inflammasome activation.

AMENDED AS SUGGESTED

Page 1

The use of electrospun (e-spun) fibers, as a functionalized system for controlled drug delivery, have numerous advantages compared to conventional dosage forms such as improved therapeutic effect, reduced toxicity, convenience, and so on [1,2].

The use of electrospun (e-spun) fibers as a functionalized system for controlled drug delivery has numerous advantages compared to conventional dosage forms, such as improved therapeutic effect, reduced toxicity, convenience and so on [1,2].

CORRECTED AS SUGGESTED BY REVIEWER

Page 1

Poly-L-lactic acid (PLA), one of the most versatile and widely tested material that has been approved by the US Food and Drug Administration (FDA) for direct contact also with biological fluids [3].

This set of words is not a sentence in English.

THIS SENTENCE HAS BEEN REVISED AND APPROPRIATELY CHANGED.

Page 2

The possibility of large scale productions combined with the simplicity of the process makes this technique very attractive for many different applications [9].

The possibility of large scale productions combined with the simplicity of the process makes this technique to be very attractive for many different applications [9].

AMENDED AS SUGGESTED BY THE REVIEWER

Page 2

Electrospinning assembly can be modified in different ways for combining materials properties.[6] So, there is an ever increasing interest in this field of electrospinning because of the useful properties of e-spun fibers which are suitable for biomedical applications [5,6].

Electrospinning assembly can be modified in different ways for combining material properties.[6] So, there is an ever increasing interest in this field of electrospinning because of useful properties of the e-spun fibers, which are suitable for biomedical applications [5,6].

MODIFIED

Page 2

The yellow colour of turmeric is mainly due to the presence of curcuminoids which are hydrophobic polyphenols, with poor solubility in water. The major curcuminoids are curcumin, demethoxycurcumin (DMC), bisdemethoxycurcumin and the most recently discovered, cyclocurcumin (CYC) also called curcumin-I, curcumin-II, curcumin-III and curcumin IV respectively [15].

The yellow color of turmeric is mainly due to the presence of curcuminoids, which are hydrophobic polyphenols with poor solubility in water. The major curcuminoids are curcumin, demethoxycurcumin (DMC), bisdemethoxycurcumin and the most recently discovered cyclocurcumin (CYC) also called curcumin-I, curcumin-II, curcumin-III and curcumin IV, respectively [15].

COMMAS ADDED AS REQUESTED

Page 2

However, the term “curcumin” is confusingly also used in the literature and commercial applications, to describe the curcumin extract that contains all four curcuminoids (I, II, III and IV) at different concentrations [16,17].

However, the term “curcumin” is confusingly also used in the literature and commercial applications to describe the curcumin extract that contains all four curcuminoids (I, II, III and IV) at different concentrations [16,17].

COMMA REMOVED AS SUGGESTED BY THE REVIEWER

Page 2

BDMC shows higher antimetastasis potency than cur-I [18,19] and has an inhibitory activity significantly greater than that of cur-I and cur-II over some leukemia cell lines (K562, KBM-5) [20] and MSK1 protein which promotes the synthesis of inflammatory cytokines [16].

Please introduce all abbreviations with higher accuracy and with clear relation to full terms.

MODIFIED AS REQUESTED

Page 2

In contraposition, DMC and Cur-I have been found to possess one and two methoxy groups respectively [19,23,24].

Contrary to that, DMC and Cur-I were found to possess one and two methoxy groups, respectively [19,23,24].

CORRECTED AS SUGGESTED BY REVIEWER

Page 2

About rates of alkaline degradation, BDMC values are lower than Cur-1 and DMC. Cur-I is stable at acid pH because of the structure of conjugated diene. In neutral or basic conditions, the phenolic OH is deprotonated and it becomes unstable [25].

As to the rates of alkaline degradation, BDMC values are lower than Cur-1 and DMC. Cur-I is stable at acid pH because of the conjugated diene structure. In neutral or basic conditions, the phenolic OH is deprotonated, and it becomes unstable [25].

AMENDED AS SUGGESTED BY REVIEWER

Page 2

Recently, efforts have been made to develop an effective drug delivery system of curcumin based on electrospinning technique for biomedical applications [26–28].

Recently, efforts were made to develop an effective drug delivery system of curcumin based on electrospinning technique for biomedical applications [26–28].

AMENDED AS PROPOSED BY REVIEWER

Page 2

It is known that after PNI inflammatory events occur. Inflammation is a defense mechanism of the organism against harmful stimuli such as infection, antigen challenge or tissue injury [31].

It is known that, after PNI, inflammatory events occur. Inflammation is a defense mechanism of the organism against harmful stimuli, such as infection, antigen challenge or tissue injury [31].

COMMAS ADDED AS SUGGESTED

Page 2

Moderate inflammatory response is key to host defense and helps to repair damaged tissues.

Moderate inflammatory response is a key to host defense and it helps to repair damaged tissues.

CORRECTED

Page 2

The most extensively studied is the NLRP3 inflammasome or cryopyrin which is formed by the cytosolic sensor molecule NLRP3, the adaptor protein ASC, and the effector molecule pro-caspase-1 [34,35].

The most extensively studied is the NLRP3 inflammasome or cryopyrin, which is formed by the cytosolic sensor molecule NLRP3, the adaptor protein ASC and the effector molecule pro-caspase-1 [34,35].

COMMAS HAS BEEN CHANGED AS SUGGESTED BY THE REVIEWER

Page 2

Lipopolysaccharide (LPS), founded in outer membrane of Gram negative bacteria can act in the first step of priming.

Lipopolysaccharide (LPS) founded in outer membrane of Gram negative bacteria can act in the first step of priming.

COMMA DELETED AS SUGGESTED

Page 2

LPS induces inflammation processes via toll-like receptor 4, and consequently, NF-kβ is activated by phosphorylation leading to the transcription of pro-inflammatory cytokines genes such as tumour necrosis factor alpha (TNF-α) and interleukin 1β (IL-1β).

LPS induces inflammation processes via toll-like receptor 4, and, consequently, NF-kβ is activated by phosphorylation leading to the transcription of pro-inflammatory cytokines genes such as tumour necrosis factor alpha (TNF-α) and interleukin 1β (IL-1β).

COMMA HAS BEEN ADDED AS PROPOSED

Page 2

ATP present in living cells is released from dying and stressed cells and may act as a and damage-associated molecular pattern (DAMP) [38].

ATP present in living cells is released from dying and stressed cells and may act as a damage-associated molecular pattern (DAMP) [38].

CORRECTED AS REQUESTED. THANK YOU

Page 2

The purpose of this article is to obtain a BDMC release system and study the effects of this curcuminoid in isolation.

The purpose of this work is to obtain a BDMC release system and study the effects of this curcuminoid in isolation.

AMENDED AS SUGGESTED BY THE REVIEWER

Page 3

Towards this aim, we fabricated e-spun PLA-BDMC loaded fibers with the purpose to obtain a functionalized system for controlled drug delivery.

Towards this aim, we fabricated e-spun PLA-BDMC loaded fibers to obtain a functionalized system for controlled drug delivery.

REWRITTEN AS PROPOSED

Page 3

To obtain PLA e-spun fibers we follow these steps.

To obtain PLA e-spun fibers, the following steps were implemented.

AMENDED AS SUGGESTED BY THE REVIEWER

Page 3

Samples were coated with platinum as electrically conductive material.

Samples were coated with platinum as an electrically conductive material.

THANK YOU, IT HAS BEEN CORRECTED.

Page 4

A DSC analysis of membranes was performed using a DSC 8000 (Perkin Elmer) equipped with Intracooler.

DSC analysis of membranes was performed using a DSC 8000 (Perkin Elmer) equipped with Intracooler.

AMENDED AS SUGGESTED BY THE REVIEWER.

Page 4

Samples were exposed to a heating cycle in the range from 40 ºC to 190 ºC with a heating rate of 20 °C/min under the atmosphere of nitrogen.

Samples were exposed to a heating cycle in the range from 40 to 190 ºC with a heating rate of 20 °C/min under the nitrogen atmosphere.

MODIFIED AS SUGGESTED

Page 4

As a result, were obtained thermograms representing the mass loss of the sample as a function of temperature.

As a result, thermograms were obtained, which are representing the mass loss as a function of temperature.

THE SENTENCE HAS BEEN REWRITTEN

Page 4

Results were calculated according to the following equation:

The results were calculated according to the following equation:

AMENDED AS PROPOSED BY THE REVIEWER

Page 4

where X represents mass fraction, R is % of weight loss of the sample; P is % of weight loss of the PLA sample; B is % of weight loss of the BDMC sample.

where X represents the mass fraction, R is the percentage of weight loss of the sample; P is the percentage of weight loss of the PLA sample; B is the percentage of weight loss of the BDMC sample.

CORRECTED AS REQUESTED

Page 4

Aliquots of each membrane were meassured by absorbance at fixed intervals of time (10, 20, 30 and 40 minuts) using an ultraviolet-visible (UV–Vis) spectrometer at excitation of 426 nm.

Aliquots of each membrane were measured by absorbance at fixed intervals of time (10, 20, 30 and 40 min) at the excitation of 426 nm using an ultraviolet-visible (UV–Vis) spectrometer.

AMENDED AS SUGGESTED BY THE REVIEWER

Page 4

A calibration curve was stablished relating weigh of BDMC with absorbance to determine the amount of BDMC incorporated in the membrane.

A calibration curve was stablished relating weight of BDMC with absorbance to determine the amount of BDMC incorporated in the membrane.

CORRECTED. THANK YOU

Page 4

The values of the same sample were added in order to calculate the total amount of BDMC loaded in the membrane.

This sentence is less clear.

THE SENTENCE HAS BEEN REWRITTEN. THANK YOU FOR DRAWING OUR ATTENTION TO THIS.

Page 4

In vitro drug release of the drug from membranes was evaluated by the dissolution technique.

In vitro drug release from membranes was evaluated by the dissolution technique.

CORRECTED AS REQUESTED

Page 4

PBST was replaced with new after each measurement.

New PBST solution was employed for each measurement.

MODIFIED AS SUGGESTED

Page 5

PLA/BDMC composite membranes autofluorescence, due to the presence of curcuminoid, was removed washing with decreased concentrations of DMSO until they lost their characteristic yellow color and rehydrated.

The PLA/BDMC composite membranes autofluorescence, due to the presence of curcuminoid, was removed by washing with decreased concentrations of DMSO until they lost their characteristic yellow color and rehydrated.

AMENDED AS PROPOSED BY THE REVIEWER

Page 5

At 24 hours the percentage of FBS in culture media was decreased to 1 %.

At 24 h, the percentage of FBS in culture media was decreased to 1 %.

CHANGED AS PROPOSED

Page 6

Figure 1 (A-C) shows the macroscopic aspect of three representative e-spun membranes fabricated.

Figure 1 (A-C) shows the macroscopic aspect of three representative fabricated e-spun membranes.

AMENDED AS SUGGESTED

Page 6

Morphology of these e-spun fibers was characterized and shown in Figure 1 (D-I).

Morphology of these e-spun fibers was characterized, and it is shown in Figure 1 (D-I).

MODIFIED AS PROPOSED BY THE REVIEWER

Page 6

The mean diameter of the PLA nanofibers was estimated to be 0,84±0,28 μm, while for PLA/BDMC composite membranes were 0,35±0,13 μm and 0,45±0,17 μm.

The mean diameter of the PLA nanofibers was estimated to be 0.84±0.28 μm, while for PLA/BDMC composite membranes were 0.35±0.13 and 0.45±0.17 μm.

DECIMALS HAVE BEEN CORRECTED. THANK YOU.

Page 6

The average of PLA fibers diameter doubled those of PLA/BDMC composite membranes suggesting that the amount of BDMC present in the electrospinning solution, had a significant effect on the final diameter of the electrospun fibers.

The average of PLA fibers diameter doubled those of PLA/BDMC composite membranes suggesting that the amount of BDMC present in the electrospinning solution had a significant effect on the final diameter of the electrospun fibers.

COMMA DELETED AS PROPOSED BY THE REVIEWER

Page 8

The DSC thermograms of the B1, B2, PLA fibers and BDMC were depicted in Figure 3.

The DSC thermograms of the B1, B2, PLA fibers and BDMC were shown in Figure 3.

AMENDED AS SUGGESTED

Page 8

The comparison of the curves in Figure 3 reveals that the electrospun smaples possessed, as produced, a much smaller crystalline phase than the commercial PLA, this being due to the rapid evaporation of the solvent upon electrospinning, that impedes crystallization.

The comparison of the curves in Figure 3 reveals that the electrospun smaples possessed, as produced, a much smaller crystalline phase than the commercial PLA, and this being due to the rapid evaporation of the solvent upon electrospinning that impedes crystallization.

MODIFIED AS SUGGESTED BY THE REVIEWER

Page 9

From the thermograms of Figure 4 it can be seen that the char residue of BDMC was much higher than that of the others, which should be due to the phenyl rings in BDMC.

From the thermograms given in Figure 4 it can be seen that the char residue of BDMC was much higher than that of the others, which should be due to the phenyl rings in BDMC.

AMENDED AS SUGGESTED

Page 9

BDMC had higher temperature of maximum decomposition and even if the temperature increased 100 °C, the BDMC did not degrade more.

BDMC had higher temperature of maximum decomposition, and, even if the temperature increased to 100 °C, the BDMC did not degrade more.

COMMAS ADDED AS PROPOSED

Page 9

If we analyzed the percentage weight loss at a concret temperature as 570 °C, it was clearly observed that the residual mass increased in the BDMC-loaded PLA fibers in a directly proportional way to the amount of BDMC they had.

If we analyzed the percentage weight loss at a selected temperature 570 °C, it was clearly observed that the residual mass increased in the BDMC-loaded PLA fibers in a directly proportional way to the amount of BDMC they had.

CHANGED AS SUGGESTED BY THE REVIEWER

Page 10

Figure 4. TGA thermograms of BDMC, PLLA fibers and BDMC-loaded PLA fibers.

Decimal points should be used instead of comma.

THANK YOU, THEY HAVE BEEN CORRECTED.

Page 10

The kinetics of release of the drug from the e-spun devices showed that the maximum value of BDMC in the medium was obtained in the first day. The maximum release of the drug was observed within first 24 hours for both, B1 and B2 membranes, where more than 80% of the BDMC was eluted to the medium.

The kinetics of the drug release from the e-spun devices showed that the maximum value of BDMC in the medium was obtained in the first day. The maximum release of the drug was observed within first 24 h for both, B1 and B2 membranes, where more than 80% of the BDMC was eluted to the medium.

AMENDED AS REQUESTED

Page 10

In this case, considering drug loading efficiency, the percentages of BDMC released in the first 24 h decreased to 41 % and 11 % for B1 and B2 respectively.

In this case, considering drug loading efficiency, the percentages of BDMC released in the first 24 h decreased to 41 and 11 % for B1 and B2, respectively.

AMENDED AS SUGGESTED BY THE REVIEWER

Page 10

This difference between B1 and B2 became smaller if we took as a reference the 351 calculated theoretical values of BDMC loaded per disk, 50 μg and 100 μg.Table 2.

Table 2 is not found in the paper.

THANK YOU. IT HAS BEEN ELIMINATED.

Figure 6. Drug release in vitro: shows the release profile of the drug at different time intervals in  PBST buffer.

Figure 5 should be considered after Figure 4.

FIGURE NUMBERS HAVE BEEN REVISED AND APPROPRIATELY CHANGED.

Page 11

The MTS test was performed in SCs cells to evaluate the cell viability of the e-spun PLA/BDMC loaded membranes. The absorbance, at 490 nm wavelength, was directly proportional to the number of viable cells (Figure 7). According to the results derived from figure 9. we can see how the B1 effect increased the absorbance values respect to the control on 1st and 3rd days, whereas this difference was not significative on day 7th.

Figure 6 is not mentioned in the text. Figure 8 should be considered before Figure 9.

FIGURE NUMBERS HAVE BEEN REVISED AND APPROPRIATELY CHANGED.

Page 11

However, on day 1st the significant difference was found when comparing the two groups with BDMC between themselves.

However, on day 1st, the significant difference was found when comparing the two groups with BDMC between themselves.

AMENDED AS SUGGESTED

Page 11

That morphological alteration would be more pronounced as the concentration of BDMC increases in the membrane.

The morphological alteration would be more pronounced as the concentration of BDMC increases in the membrane.

MODIFIED AS PROPOSED BY THE REVIEWER

Page 12

After 24 h SCs were stimulated in two steps with 396 LPS and ATP inducing inflammatory response (Figure 9).

After 24 h, SCs were stimulated in two steps with 396 LPS and ATP inducing inflammatory response (Figure 9).

COMMA ADDED AS SUGGESTED

Page 12

The results demonstrated that under pro-inflammatory conditions the expression of NF-kβ was significantly reduced in presence of BDMC when compared to other groups. Moreover, TNFα decreased very significantly in the B1 and B2 membranes.

Components of the inflammasome complex, NLRP3 and Casp1, had lower expression levels when cells were seeded on membranes loaded with BDMC. Results also showed IL-18 mRNA expression was decreased in presence of BDMC in a highly significative way.

The results demonstrated that, under pro-inflammatory conditions, the expression of NF-kβ was significantly reduced in presence of BDMC when compared to other groups. Moreover, TNFα decreased very significantly in the B1 and B2 membranes.

The components of the inflammasome complex, NLRP3 and Casp1, had lower expression levels when cells were seeded on membranes loaded with BDMC. The results also showed IL-18 mRNA expression was decreased in presence of BDMC in a highly significant way.

MODIFIED AS RECOMMENDED

Page 12

Consistently, results from RT-qPCR indicated that addition of BDMC to the e-spun membranes had suppressive effects on inflammasome activation, assembly and subsequent release of mature form of IL-18 compared to the control in a no dose-depedent manner.

Consistently, the results from RT-qPCR indicated that addition of BDMC to the e-spun membranes had suppressive effects on inflammasome activation, assembly and subsequent release of mature form of IL-18 compared to the control in a no dose-dependent manner.

CORRECTED. THANK YOU

Page 13

Protein expression profiles showed that membranes loaded with BDMC reduced NLRP3 inflammasome related proteins.

The protein expression profiles showed that membranes loaded with BDMC reduced NLRP3 inflammasome related proteins.

AMENDED AS PROPOSED BY THE REVIEWER

Page 14

TE is a scientific research area that has been a massive increasing interest in recent years for an effective approach to the repair of tissues [40].

TE is a scientific research area with a massive increasing interest in recent years for an effective approach to the tissue repair [40].

THE SENTENCE HAS BEEN REWRITTEN

Page 14

However, not changes were observed in their surfaces without compromising bulk properties of biomaterial.

However, no changes were observed in their surfaces without compromising bulk properties of biomaterial.

CORRECTED, THANK YOU.

Page 14

However, the increased diameter of B2 fibers respect to B1 may be the result of variations in the electrospinning solution properties such as viscosity due to differences in drug concentration. Studies have revealed that an increase of viscosity may be promoved by an increase of a BDMC structural analogue, curcumin [45].

However, the increased diameter of B2 fibers in reference to B1 may be the result of variations in the electrospinning solution properties, such as viscosity due to differences in drug concentration. The studies revealed that an increase of viscosity may be promoted by an increase of a BDMC structural analogue, curcumin [45].

AMENDED AS SUGGESTED BY THE REVIEWER

Page 15

One day after treatment, B1 membranes significantly showed the highest proliferative effect of the drug.

One day after treatment, B1 membranes showed the highest proliferative effect of the drug.

MODIFIED AS SUGGESTED

Page 15

Drug released was not enough to have an effect on the cells.

The drug release was not enough to have an effect on the cells.

AMENDED AS PROPOSED

Page 15

It is known that the effect of the aligned substrate is significant, on axon and glial cell orientation and maduration [47–49].

It is known that the effect of the aligned substrate is significant on axon and glial cell orientation and maduration [47–49].

COMMA DELETED

What is a “maduration”?

IS THE PROCESS THAT ALLOWS A CELL TO BECOME FUNCTIONAL AND SPECIALIZED. THE DEVELOPMENTAL STAGES OF CELLS.

FOR EXAMPLE: SC DEVELOPMENT STARTS FROM NEURAL CREST CELLS, THROUGH SC PRECURSORS, IMMATURE SCS, PRO-MYELINATING SCS TO MATURE MYELINATING AND/OR NON-MYELINATING SCS.

Bibliography “Schwann cell development, maturation and regeneration: a focus on classic and emerging intracellular signaling pathways” Neural Regen Res. 2017 Jul; 12(7): 1013–1023. doi: 10.4103/1673-5374.211172

Page 15

They are promising drug candidates for the treatment of many diseases but their use in therapeutic treatments has been hampered by low aqueous solubility and chemical instability [53].

They are promising drug candidates for the treatment of many diseases, but their use in therapeutic treatments is hampered by low aqueous solubility and chemical instability [53].

AMENDED AS SUGGESTED BY THE REVIEWER

Page 16

In the present study we found that in proinflammatory conditions, BDMC acted in a similar way to its analagous curcumin.

In the present study, we found that, in proinflammatory conditions, BDMC acted in a similar way to its analogous curcumin.

COMMAS ADDED AS SUGGESTED

WE WOULD LIKE TO THANK THE REVIEWER FOR HIS/HER HELP AND SUGGESTIONS TO CLARIFY SOME PARTS OF THE TEXT. ALL THE CORRECTIONS HAVE BEEN MADE.

Please find enclosed the new electronic version of the paper entitled “Characterization of electrospun BDMC loaded PLA nanofibers with drug delivery function and anti-inflammatory activity” by Maria José Morillo-Bargues, Andrea Olivos Osorno, Consuelo Guerri, Manuel Monleón Pradas and Cristina Martínez-Ramos, submitted to be considered for publication in International Journal of Molecular Sciences.

Yours sincerely,

Cristina Martínez Ramos

Centre for Biomaterials and Tissue Engineering. Universidad Politécnica de Valencia

46022 Valencia, Spain

Email: cris_mr_1980@hotmail.com

Round 2

Reviewer 2 Report

All of the issues have been addressed.